# Synaptotagmin-11 facilitates assembly of a presynaptic signaling complex in post-Golgi cargo vesicles

Luca Trovò [1], Stylianos Kouvaros [1], Jochen Schwenk[2], Diego Fernandez-Fernandez [1], Thorsten Fritzius [1], Pascal Dominic Rem [1], Simon Früh [1], Martin Gassmann [1], Bernd Fakler [2,3,4], Josef Bischofberger[1] & Bernhard Bettler [1]✉

## Abstract

GABA$_B$ receptors (GBRs), the G protein-coupled receptors for GABA, regulate synaptic transmission throughout the brain. A main synaptic function of GBRs is the gating of Cav2.2-type Ca$^{2+}$ channels. However, the cellular compartment where stable GBR/Cav2.2 signaling complexes form remains unknown. In this study, we demonstrate that the vesicular protein synaptotagmin-11 (Syt11) binds to both the auxiliary GBR subunit KCTD16 and Cav2.2 channels. Through these dual interactions, Syt11 recruits GBRs and Cav2.2 channels to post-Golgi vesicles, thus facilitating assembly of GBR/Cav2.2 signaling complexes. In addition, Syt11 stabilizes GBRs and Cav2.2 channels at the neuronal plasma membrane by inhibiting constitutive internalization. Neurons of Syt11 knockout mice exhibit deficits in presynaptic GBRs and Cav2.2 channels, reduced neurotransmitter release, and decreased GBR-mediated presynaptic inhibition, highlighting the critical role of Syt11 in the assembly and stable expression of GBR/Cav2.2 complexes. These findings support that Syt11 acts as a vesicular scaffold protein, aiding in the assembly of signaling complexes from low-abundance components within transport vesicles. This mechanism enables insertion of pre-assembled functional signaling units into the synaptic membrane.

**Keywords** GABA-B; KCTD16; Cav2.2; Transport Vesicle; Receptor Signaling Complex Assembly

**Subject Categories** Membranes & Trafficking; Neuroscience

## Introduction

GBRs are G protein-coupled receptors (GPCRs) that regulate synaptic transmission by controlling the activity of voltage-sensitive Ca$^{2+}$ (Cav) channels and inwardly-rectifying Kir3-type K$^+$ channels through the Gβγ subunit of the activated G protein (Gassmann and Bettler, 2012; Padgett and Slesinger, 2010; Rose and Wickman, 2022; Turecek et al, 2014). To ensure rapid and specific channel gating, GBRs must be in close spatial proximity to effector channels (Ferre et al, 2022). Although GPCRs can assemble with the heterotrimeric G protein before incorporation into the plasma membrane (David et al, 2006; Dupre et al, 2006), the cellular compartment within the biogenic pathway where GPCRs and effector channels assemble into signaling-competent complexes remains unclear.

The heterodimeric GB1a/2 and GB1b/2 receptor subtypes localize to pre- and postsynaptic membranes, respectively (Vigot et al, 2006). Receptor-associated KCTD8, -12, -12b, and -16 proteins, which interact with the G protein Gβγ subunits, regulate the kinetics of GBR-gated currents (Bhandari et al, 2021; Fritzius and Bettler, 2020; Fritzius et al, 2024; Fritzius et al, 2017; Schwenk et al, 2010; Schwenk et al, 2016; Turecek et al, 2014; Zheng et al, 2019; Zuo et al, 2019). Proteomic studies suggested that KCTD16 also acts as a scaffold protein that links GBRs with effector Cav2.2 channels (Schwenk et al, 2016). The KCTD proteins associate with heterodimeric GBRs at the cytosolic side of the ER membrane (Ivankova et al, 2013), indicating that Cav2.2 channels could interact with GBRs early during biogenesis, and that pre-assembled GBR/Cav2.2 signaling complexes are subsequently transported to their functional sites. Transport of GB1a/2 receptors along microtubules in post-Golgi vesicles requires binding of the receptor to the amyloid precursor protein (APP) (Biermann et al, 2010; Dinamarca et al, 2019; Maday et al, 2014). In addition, our proteomic analysis of native GBRs revealed that Syt11, a synaptotagmin isoform residing in mobile vesicles in neurons (Dean et al, 2012a; Shimojo et al, 2019), directly or indirectly associates with GB1a/2 receptors (Dinamarca et al, 2019; Schwenk et al, 2016). Moreover, GB1a/2 receptor complexes purified from $APP^{-/-}$ brains contain significantly less Syt11 protein (Dinamarca et al, 2019), and both Syt11 and GBRs were identified as constituents of APP complexes affinity purified from brain membranes (Norstrom et al, 2010). Therefore, proteomic data

[1]Department of Biomedicine, University of Basel, Basel, Switzerland. [2]Institute of Physiology, Faculty of Medicine, University of Freiburg, Freiburg, Germany. [3]CIBSS Center for Integrative Biological Signalling Studies, University of Freiburg, Freiburg, Germany. [4]Center for Basics in NeuroModulation, Freiburg, Germany. ✉E-mail: bernhard.bettler@unibas.ch

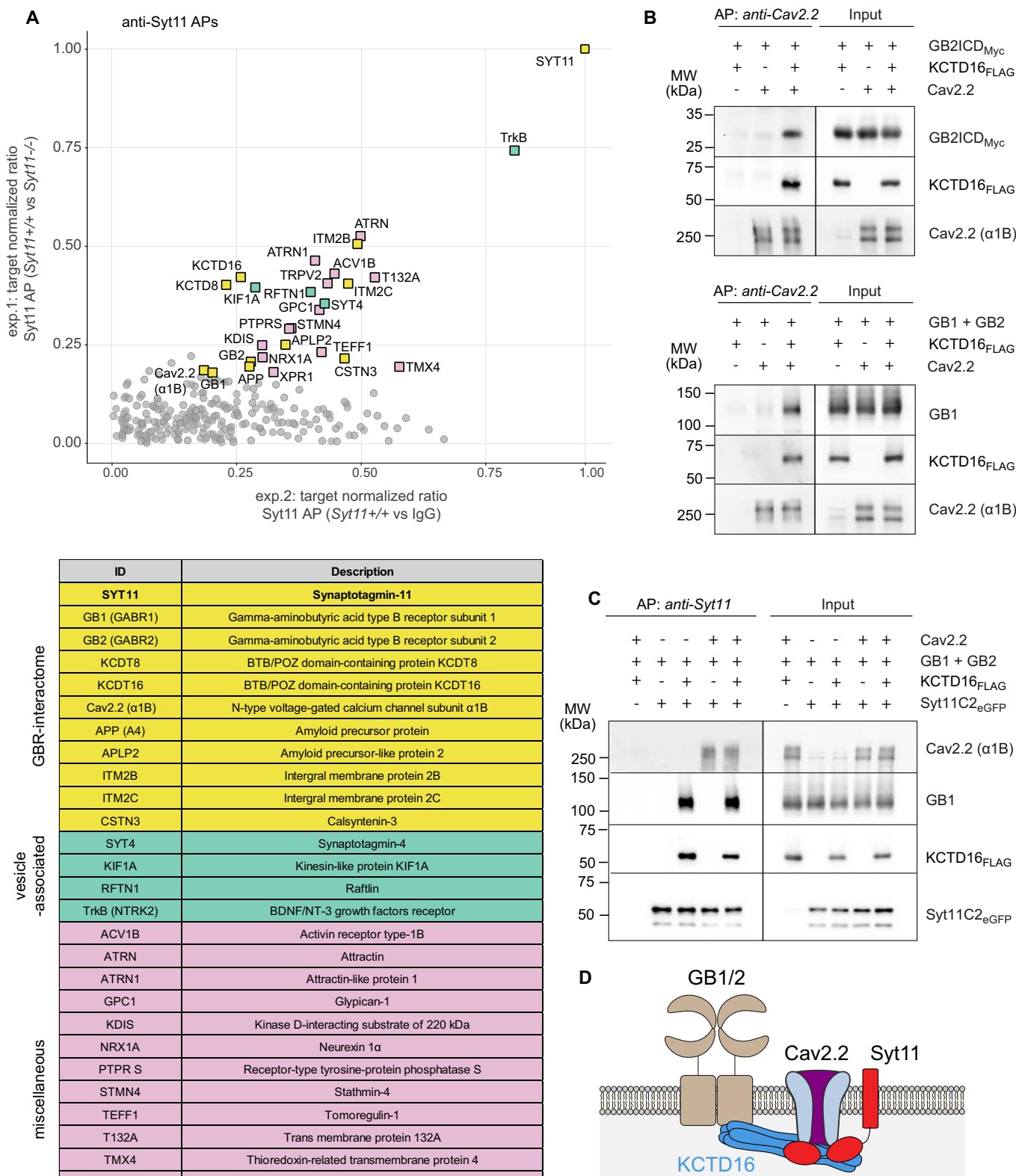

**A** anti-Syt11 APs

**B**

**C**

**D**

◀ **Figure 1. Syt11 assembles with GBR/Cav2.2 complexes by binding to both KCTD16 and Cav2.2.**

(A) Abundance ratios of proteins identified in APs from solubilized *Syt11*$^{+/+}$ and *Syt11*$^{-/-}$ mouse brain membranes using target-specific anti-Syt11 antibodies and control IgG (target-normalized ratio values (tnRs), see Methods). Proteins that were consistently enriched (tnR > 0.18) in Syt11 APs from *Syt11*$^{+/+}$ brain membranes compared to control Syt11 APs from *Syt11*$^{-/-}$ brain membranes (exp.1) and control APs with pre-immunization IgG (exp.2) are indicated and listed in the table. Among the identified 28 proteins, 11 were previously identified as constituents of native GBR-complexes (highlighted in yellow), 4 are vesicle-associated proteins (green), and 13 miscellaneous proteins (pink). (B, C) AP experiments in transfected HEK293T cells. (B) Co-AP of the Myc-tagged intracellular C-terminal domain of GB2 (GB2ICD; top), which mediates binding of GBRs to KCTD proteins (Schwenk et al, 2010) or GB1 (bottom) with Cav2.2 in the presence of FLAG-tagged KCTD16. (C) Co-AP of GB1 and Cav2.2 with the eGFP-tagged cytoplasmic C2A and C2B domains of Syt11 fused to eGFP (Syt11C2$_{eGFP}$) in the presence of FLAG-tagged KCTD16. Notably, Cav2.2, but not GB1 is detected in anti-Syt11C2$_{eGFP}$ APs in the absence of KCTD16, indicating that Syt11 directly interacts with Cav2.2. The α1B subunit of Cav2.2 channels, co-expressed with auxiliary β and α2δ subunits, was identified on Western blots using the anti-Cav2.2 antibody from Millipore (# AB5154). For APs demonstrating the specificity of the interaction of KCTD16 with Syt11, see Fig. EV1B. (D) Scheme depicting the interaction of Syt11 with GBR/Cav2.2 complexes. Syt11 directly binds to Cav2.2 and KCTD16, an auxiliary subunit of GBRs that links the GB2 subunit of GBRs to Cav2.2 channels. Source data are available online for this figure.

support the presence of Syt11 in APP/GB1a/2 complexes within post-Golgi vesicles (Dinamarca et al, 2019). However, it remained unknown how Syt11 interacts with GBRs and whether Syt11 influences vesicular transport of GBRs.

Syt family members are type I membrane proteins known to regulate exocytosis, endocytosis, and vesicle trafficking (Südhof and Rizo, 1996; Wolfes and Dean, 2020). They possess cytoplasmic C2A and C2B domains and act as Ca$^{2+}$ sensors for SNARE-dependent vesicle fusion (Brose, 2008; Schneggenburger and Rosenmund, 2015; Südhof, 2013). However, both Syt11 and the related Syt4 exhibit reduced affinity for Ca$^{2+}$ due to structural alterations in their C2 domains (Dai et al, 2004; von Poser et al, 1997). Syt11 is an integral vesicle protein that inhibits clathrin-mediated and bulk endocytosis in dorsal root ganglion neurons (Wang et al, 2016). Therefore, Syt11 may not only play a role in vesicular GBR transport but also in inhibiting internalization of receptors from the cell surface (Benke et al, 2015). Ablation of Syt11 in excitatory forebrain neurons impaired synaptic plasticity and memory formation, indicating that Syt11 influences synaptic transmission (Shimojo et al, 2019).

In the present study, we show that Syt11 is a vesicular scaffold protein that recruits individual components of multi-protein signaling complexes into post-Golgi transport vesicles, which facilitates assembly of the signaling complex. Specifically, we show that Syt11 interacts with Cav2.2 channels and GBRs in post-Golgi vesicles and increases cell-surface stability of GBR/Cav2.2 complexes by reducing endocytosis and lysosomal degradation. Consistent with these findings, *Syt11*$^{-/-}$ neurons exhibit deficits in presynaptic GBRs and Cav2.2 channels, neurotransmitter release, and GBR-mediated presynaptic inhibition.

## Results

### Syt11 interactions with KCTD16 and Cav2.2 form GBR/Cav2.2 complexes

Syt11 is a constituent of native GBR complexes (Dinamarca et al, 2019; Schwenk et al, 2016). The binding partners of Syt11 within the GBR complex, however, are unknown, as are its possible effects on receptor trafficking and/or signaling. Therefore, we performed affinity purifications (APs) with anti-Syt11 antibodies, reverse to our previous anti-GB1/anti-GB2 APs (Schwenk et al, 2016), to determine the composition of Syt11 complexes in adult mouse brain membranes. To control the APs, we generated *Syt11*$^{-/-}$ mice

using CRISPR/Cas9. Since *Syt11*$^{-/-}$ mice exhibited perinatal lethality, as previously described (Shimojo et al, 2019), we used *Syt11*$^{-/-}$ brains collected at postnatal day 5 (P5) as controls. Finally, we identified 28 proteins consistently enriched in anti-Syt11 APs compared to APs with pre-immunization IgG or APs with anti-Syt11 from *Syt11*$^{-/-}$ brain membranes (Fig. 1A, abundance ratio plot and table). Ten of the Syt11-interacting proteins were previously identified in anti-GBR APs (Dinamarca et al, 2019; Schwenk et al, 2010; Schwenk et al, 2016), including GB1, GB2, KCTD16 and the Cav2.2 (N-type voltage-gated Ca$^{2+}$ channel) subunit α1B. Cav2.2 channels interact with KCTD16 (Müller et al, 2010) and require KCTD16 for co-assembly with GBR complexes (Schwenk et al, 2016), suggesting that Syt11 binds to GBR/Cav2.2 complexes through its association with KCTD16. This suggestion was confirmed in AP experiments from transfected HEK293T cells in which GBRs associate with Cav2.2 via KCTD16 (Figs. 1B and EV1A). Furthermore, the APs indicated that Syt11 interacts with GBR/Cav2.2 complexes in two ways: firstly, by binding to KCTD16, and secondly, by directly binding to Cav2.2 (Fig. 1C). KCTD8 was also identified in anti-Syt11 APs from mouse brains (Fig. 1A). However, KCTD8 failed to co-purify with Syt11 in transfected HEK293T cells (Fig. EV1B), thus indicating that KCTD8 likely co-purified with KCTD16 as a constituent of KCTD heteropentamers (Fritzius et al, 2017). Notably, the α1A subunit of Cav2.1 channels (P/Q-type voltage-gated Ca$^{2+}$ channels) failed to co-purify with Syt11 from mouse brains (Fig. 1A). In addition, the α1A subunit exhibited a significantly decreased propensity to co-purify Syt11 in APs from transfected HEK293T cells compared to the α1B subunit of Cav2.2 channels (Fig. EV1C). Overall, our proteomic analysis suggests the existence of a multi-protein complex that includes GBRs, KCTD16, Cav2.2 channels and Syt11, with Syt11 linking Cav2.2 channels to GBRs through its interaction with KCTD16 (Fig. 1D).

Binding of Syt11 to GBRs through KCTD16 could potentially influence G protein signaling of the receptor. To analyze whether Syt11 modulates Gα signaling of GBRs, we used a luciferase-reporter assay in transfected HEK293T cells (Rem et al, 2023). Cells co-expressing GBRs and KCTD16 exhibited similar sigmoidal GABA dose-response curves, irrespective of the presence of Syt11 (Fig. 2A). Luciferase activity at baseline was not significantly different in the presence or absence of Syt11, supporting that Syt11 does not influence constitutive Gα signaling of GBRs (Fig. 2A). We also addressed whether Syt11 modulates Gβγ signaling of GBRs using a BRET assay based on the binding of Venus-tagged Gβγ to a membrane-associated GRK3ct-luciferase (masGRK3ct-NanoLuc) (Hollins et al, 2009; Masuho et al,

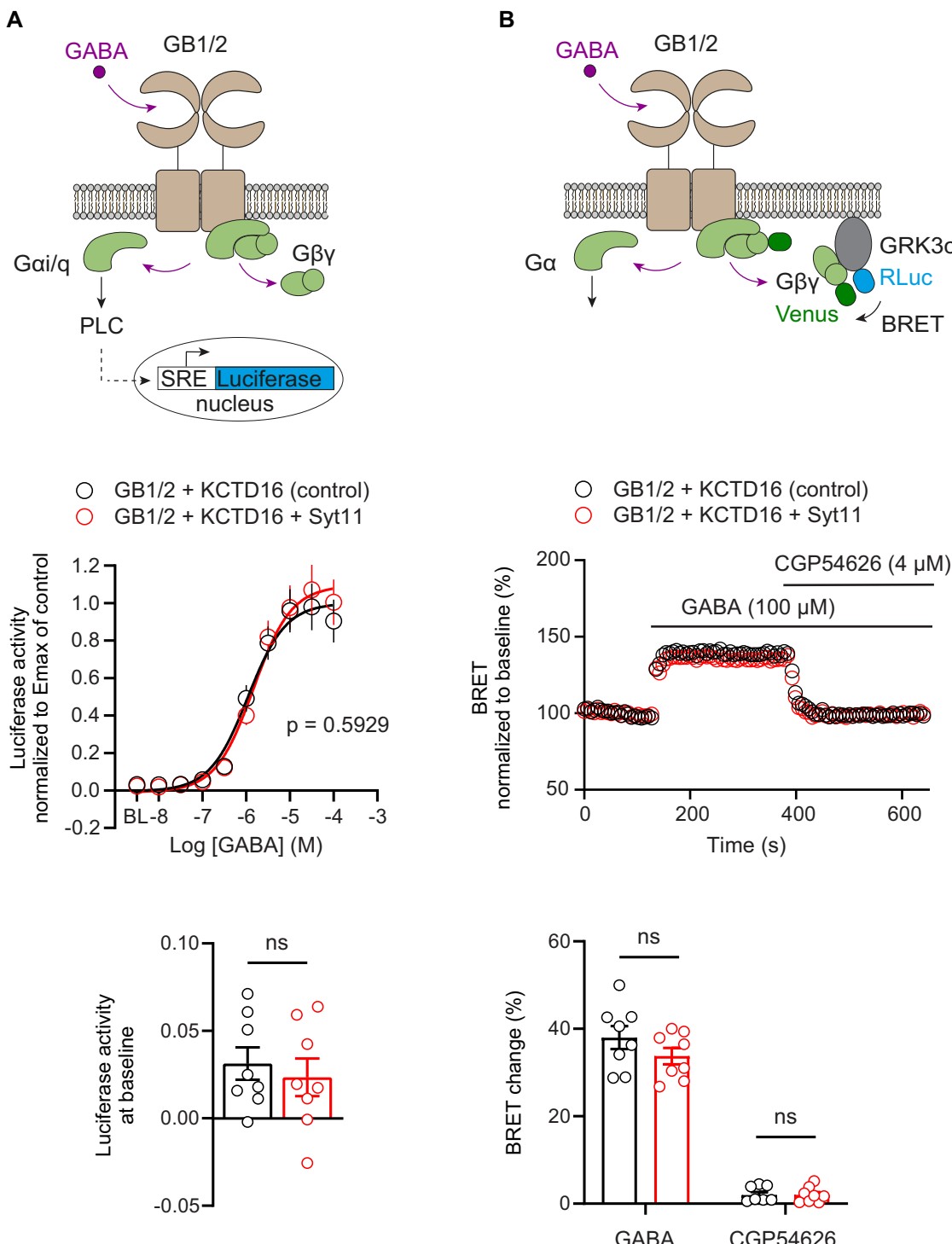

**Figure 2.  Syt11 binding to GBRs does not modulate G protein signaling.**

(**A**) GABA dose-response curves of GBRs expressed in HEK293T cells together with KCTD16 in the absence (control) or presence of Syt11. Gα signaling was monitored using a luciferase-reporter assay based on artificially coupling GBRs via chimeric Gαqi to phospholipase C (top). Expression of Syt11 does not significantly change basal and GABA-induced luciferase activity. Non-linear regression curve fits of $n = 8$ independent experiments per condition. Mean ± SEM, $p = 0.5929$, extra sum-of-squares F-test (middle). Baseline activity (BL), $p = 0.5882$, unpaired t-test (bottom). (**B**) Gβγ released upon GBR activation in HEK293T cells was monitored using a BRET assay reporting the binding of Venus-tagged Gβγ to a membrane-associated GRK3ct-luciferase (top). Representative experiments (middle) and quantification of BRET changes (bottom) induced by the application of GABA and the inverse agonist CGP54626. Co-expression of Syt11 does not significantly alter GABA ($p = 0.2068$) and CGP54626 ($p = 0.9777$) induced BRET changes compared to control (unpaired Student's t test). Mean ± SEM from $n = 8$ independent experiments recorded in triplicates, ns = not significant. Source data are available online for this figure.

2015). The BRET change induced by the application of 100 µM GABA to cells co-expressing GBRs, KCTD16, and the BRET sensors was not significantly different in the presence or absence of Syt11 (Fig. 2B). After blocking GBRs with 4 µM CGP54626, an inverse agonist inhibiting constitutive GBR activity (Grünewald et al, 2002), the BRET signal returned to baseline without undershooting, irrespective of the presence of Syt11 (Fig. 2B). The data from CGP54626 experiments corroborate that Syt11 does not induce constitutive receptor activity. Collectively, these biochemical experiments support the conclusion that Syt11 does not influence Gα or Gβγ signaling of GBRs.

## Post-Golgi vesicles transport GBR/Syt11 complexes to pre- and postsynaptic sites

Syt11-interacting proteins also included the kinesin-3 motor KIF1A (Fig. 1A), which mediates transport of various vesicles in axons and dendrites, such as dense core vesicles (DCVs) and other secretory carriers (Lo et al, 2011; Maday et al, 2014; Stucchi et al, 2018). Moreover, the presence of APP in the Syt11-interactome indicates that Syt11 is a constituent of axonal GBR-complexes that use APP for axonal receptor trafficking (Dinamarca et al, 2019). To track GBR/Syt11 complexes in living cells, we used bimolecular fluorescence complementation (BiFC) (Das et al, 2016; Dinamarca et al, 2019). Syt11 and GB2 were tagged at their C-termini with the N- and C-terminal fragments of fluorescent Venus protein (VN, VC; Fig. 3A). We only observed GB2-VC/Syt11-VN BiFC in transfected HEK293T cells expressing GB2-VC and Syt11-VN in the presence of KCTD16, corroborating that Syt11 binds to GB2 via KCTD16 (Fig. EV2A). The Venus fragment tags on GB2 and Syt11 did not impede nor enhance KCTD16-mediated complex formation, as demonstrated by co-AP of comparable amounts of Syt11 and KCTD16 with GB2 from membranes of transfected HEK293 cells, both in the absence and presence of the Venus fragment tags (Fig. EV2B). Binding of Syt11 to KCTD16 was dependent on the C2A and C2B domains of Syt11, as Syt11ΔC2-VN lacking these domains showed no BiFC with GB2-VC in the presence of KCTD16 (Fig. EV2A). Notably, we also observed no BiFC when expressing Syt1-VN instead of Syt11-VN together with GB2-VC (Fig. EV2A). Consistent with the results in HEK293T cells, the BiFC signal between GB2-VC and Syt11-VN was significantly reduced by $64.8 \pm 2.26\%$ in transfected cultured hippocampal neurons from $Kctd16^{-/-}$ compared to $Kctd16^{+/+}$ mice (Fig. 3B, $n = 23/22$ neurons, $p < 0.0001$, Mann–Whitney U test). Background BiFC in the soma of $Kctd16^{-/-}$ neurons is likely attributable to Venus self-assembly resulting from random collision in the ER (Shyu and Hu, 2008). In WT neurons, GB2-VC/Syt11-VN complexes partially co-localized with the presynaptic marker synaptophysin and the postsynaptic marker PSD95 (Fig. 3C). In dendritic spines, GB2-VC/Syt11-VN complexes were predominantly observed in spine necks, while PSD95 expression was highest in spine heads (Fig. 3D). These findings align with studies demonstrating that postsynaptic GBRs cluster at extrasynaptic sites in dendritic spines (Kulik et al, 2006).

Syt11 localizes to mobile vesicles in axons and dendrites (Shimojo et al, 2019). We used live-cell confocal imaging to analyze the mobility of GB2-VC/Syt11-VN complexes in cultured hippocampal neurons (Movie EV1). Kymographs revealed bidirectional trafficking of vesicles exhibiting Venus fluorescence in both axons and dendrites (Figs. 3E and EV3A). The average velocities of

anterograde and retrograde trafficking ranged from 1-3 µm/s (Fig. EV3B), which is consistent with fast kinesin-dependent transport along microtubules (Maday et al, 2014). To characterize the vesicles containing GB2-VC/Syt11-VN complexes, we used NPY-mCherry as a marker for Golgi-derived vesicles delivering cargo to the plasma membrane (Arora et al, 2017; Das et al, 2016), and Rab5-mCherry as a marker for recycling endosomes (Kalin et al, 2015; Rink et al, 2005) (Fig. 3F). Quantification using Manders' coefficients revealed a significantly greater degree of co-localization between GB2-VC/Syt11-VN complexes and NPY-mCherry compared to Rab5-mCherry in both axons and dendrites (Fig. EV3C). These findings indicate that the transport of GBR/Syt11 complexes primarily occurs through Golgi-derived vesicles.

## Syt11 facilitates recruitment of GBR/Cav2.2 complexes to post-Golgi vesicles

Several studies support the transport of synaptic components as pre-assembled macromolecular complexes during synapse development (Ahmari et al, 2000; Dresbach et al, 2006; Shapira et al, 2003; Zhai et al, 2001). We addressed whether Syt11 plays a role in recruiting pre-assembled GBR/Cav2.2 complexes to transport vesicles. To achieve higher resolution beyond the classical diffraction limit of optical microscopy (180–200 nm), we employed structured illumination microscopy (SIM) for visualization of individual transport vesicles (Huang et al, 2010). Cultured hippocampal neurons from $Syt11^{+/+}$ and $Syt11^{-/-}$ embryos were fixed at DIV14 and immunostained for endogenous GB2, Cav2.2, and NPY, a marker for post-Golgi-derived vesicles (Fig. 4A). While endogenous NPY protein is typically considered a marker for GABAergic neurons (Fuentealba et al, 2008; Karagiannis et al, 2009), we observed its widespread expression at moderate levels in cultured glutamatergic hippocampal neurons, consistent with a previous report (Ramamoorthy et al, 2011). Through 3D reconstruction, we were able to classify NPY+ vesicles into four distinct groups: (1) Double positive (GB2+, Cav2.2+), (2) GB2 single positive (GB2+, Cav2.2-), (3) Cav2.2 single positive (GB2-, Cav2.2+), and (4) double negative (GB2-, Cav2.2-) (Fig. 4B and Movie EV2). The density of NPY+ vesicles in both axons and dendrites showed no significant difference between $Syt11^{+/+}$ and $Syt11^{-/-}$ neurons (Fig. 4C), suggesting that the absence of Syt11 does not affect vesicle generation. However, in both the axons and dendrites of $Syt11^{-/-}$ neurons, there was a significant decrease in the proportion of NPY+ vesicles that were double positive for GB2 and Cav2.2, accompanied by an increase in the proportion of GB2 single positive and Cav2.2 single positive vesicles (Fig. 4D). Similarly, the proportion of vesicles that were double positive for GB2 and Cav2.2 within the GB2+ or Cav2.2+ vesicle populations was significantly decreased in $Syt11^{-/-}$ axons and dendrites (Fig. 4E). Collectively, these findings are consistent with Syt11 facilitating recruitment of GBR/Cav2.2 complexes to NPY+ post-Golgi transport vesicles.

## Syt11 increases GBR and Cav2.2 channel surface availability by inhibiting endocytosis

Syt11 vesicles recycle via the plasma membrane (Shimojo et al, 2019), suggesting a potential role for Syt11 in modulating surface availability of GBR/Cav2.2 complexes. In line with its reported

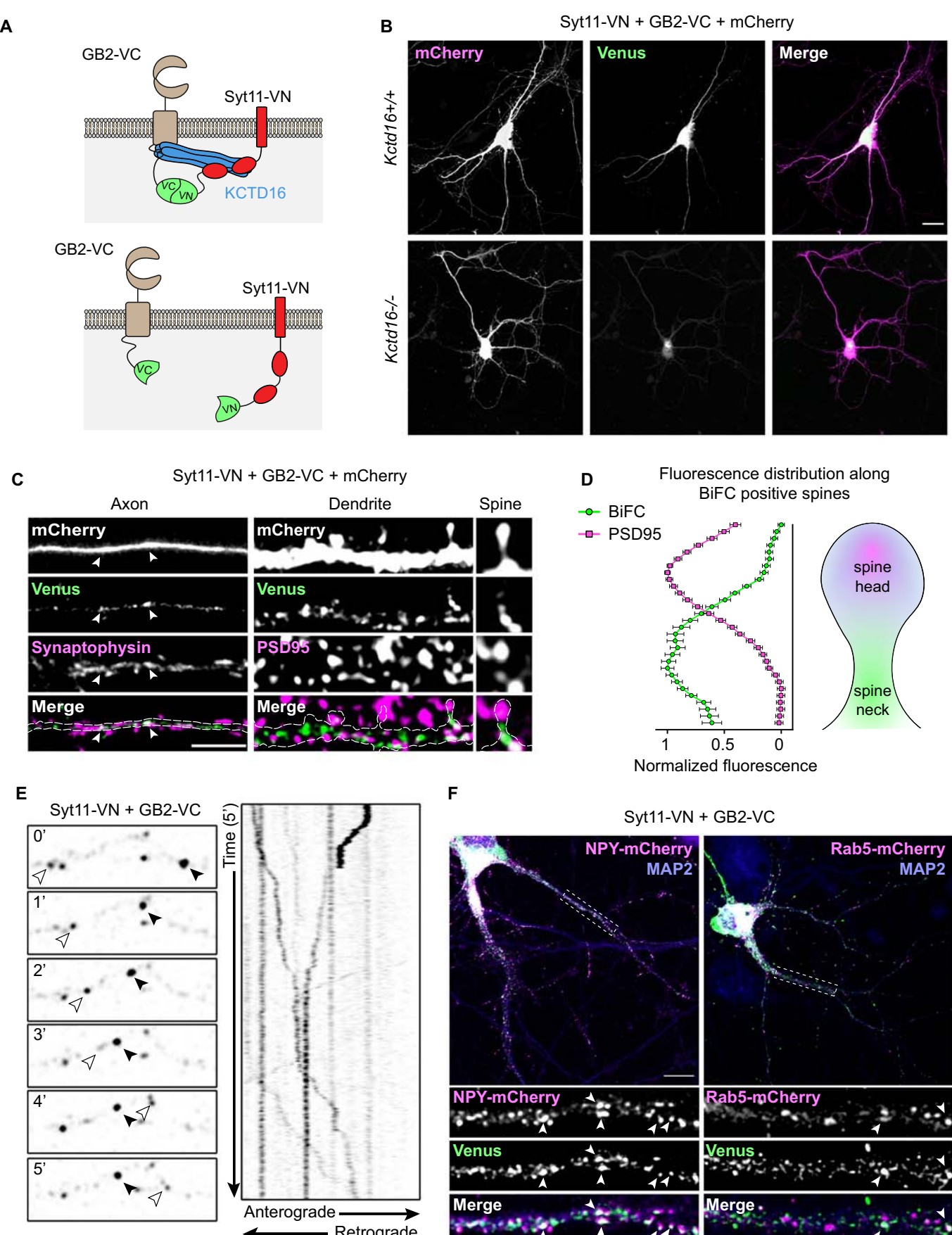

**A** GB2-VC    Syt11-VN    KCTD16

**B** Syt11-VN + GB2-VC + mCherry

mCherry    Venus    Merge

Kctd16+/+

Kctd16-/-

**C** Syt11-VN + GB2-VC + mCherry

Axon    Dendrite    Spine

mCherry    mCherry

Venus    Venus

Synaptophysin    PSD95

Merge    Merge

**D** Fluorescence distribution along BiFC positive spines

BiFC    PSD95

spine head

spine neck

1    0.5    0

Normalized fluorescence

**E** Syt11-VN + GB2-VC

0'    1'    2'    3'    4'    5'

Time (5')

Anterograde

Retrograde

**F** Syt11-VN + GB2-VC

NPY-mCherry    MAP2    Rab5-mCherry    MAP2

NPY-mCherry    Rab5-mCherry

Venus    Venus

Merge    Merge

**Figure 3. GBR/Syt11 complexes traffic in axons and dendrites and localize to synaptic sites.**

(A) Scheme illustrating the principle of BiFC. Complex formation of Syt11-VN with GB2-VC in the presence of KCTD16 leads to the reconstitution of Venus fluorescence. For validation of the GB2-VC/Syt11-VN BiFC in transfected HEK293T cells, see Fig. EV2. (B) Representative confocal images of cultured $Kctd16^{+/+}$ and $Kctd16^{-/-}$ hippocampal neurons (DIV10) transfected with Syt11-VN and GB2-VC. Venus BiFC is observed in axons and dendrites of $KCTD16^{+/+}$ neurons. In $Kctd16^{-/-}$ neurons, low background BiFC in the soma is likely due to Venus self-assembly in the ER. Transfected neurons were identified using mCherry as a volume marker. Scale bar: 10 μm. (C) Higher magnification of an axon and dendrite of a mature hippocampal neuron (DIV14) transfected with Syt11-VN, GB2-VC, and mCherry as a volume marker. The BiFC complex (Venus) partly co-localized with endogenous synaptophysin in axons (arrowheads). In dendrites, the BiFC complex localized to dendritic shafts and spine necks but not to spine heads, as identified by PSD95 staining. Scale bar: 5 μm. (D) BiFC and endogenous PSD95 fluorescence along BiFC positive spines (normalized to the peak fluorescence). The BiFC signal is high in spine necks and absent from spine heads, contrasting with the distribution of PSD95. Data are presented as mean ± SEM from $n = 83$ spines (4 independent preparations). (E) Time-lapse images and a related kymograph of well-separated fluorescent Syt11-VN/GB2-VC complexes moving anterogradely (white arrowhead) and retrogradely (black arrowhead) in an axon. For quantification of kymographs, see Fig. EV3A and B. (F) Representative confocal images of hippocampal neurons transfected with Syt11-VN, GB2-VC, and either NPY-mCherry or Rab5-mCherry. The fluorescent Syt11-VN/GB2-VC complex predominantly co-localizes with NPY-mCherry. MAP2 staining identifies dendrites. Higher magnifications of dendrites are shown at the bottom. Arrowheads indicate examples of co-localization. Scale bar: 10 μm. For quantification of co-localization, see Fig. EV3C. Source data are available online for this figure.

---

function as an endocytosis clamp (Wang et al, 2016), we observed a significant increase in the uptake of fluorophore-conjugated transferrin-AF647, a marker for early and recycling endosomes (Maxfield and McGraw, 2004), in $Syt11^{-/-}$ neurons (Fig. EV4A). Quantitative co-localization analysis with transferrin-AF647 further revealed a significant increase in the endocytosis of endogenous GB2 and Cav2.2 in $Syt11^{-/-}$ neurons (Fig. EV4A,B). In contrast, endocytosis of endogenous adenosine A1 receptors (A1Rs), which were not detected in anti-Syt11 APs (Fig. 1A), was similar between genotypes (Fig. EV4A,B). Co-localization of GB2 with LAMP1, an established lysosome marker, was increased in $Syt11^{-/-}$ neurons (Fig. EV4C), indicating augmented lysosomal degradation of GBRs (Grampp et al, 2008; Zemoura et al, 2019). Collectively, these findings indicate that Syt11 stabilizes associated GBRs and Cav2.2 channels at the neuronal plasma membrane by inhibiting endocytosis.

## Decreased release probability and reduced GBR-mediated inhibition of glutamate release in cultured hippocampal $Syt11^{-/-}$ neurons

Investigating the potential functional consequences of impaired co-trafficking and cell surface expression of GBRs and Cav2.2 channels in the absence of Syt11, we assessed excitatory synaptic transmission in cultured hippocampal neurons. We monitored spontaneous postsynaptic activity under GABA$_A$ receptor blockade with gabazine (10 μM) in $Syt11^{+/+}$ and $Syt11^{-/-}$ neurons. We observed a significant reduction in the frequency of spontaneous excitatory postsynaptic currents (sEPSCs) in $Syt11^{-/-}$ neurons compared to $Syt11^{+/+}$ neurons (Fig. 5A,B). This reduction was evident from the shift in the distribution of sEPSCs towards larger inter-event intervals (Fig. 5B), without altering sEPSC amplitudes (Fig. 5C) or kinetics (Appendix Fig. S1). Because the frequency of miniature EPSC (mEPSCs) remained unchanged (see below), this suggests a reduced probability of action-potential dependent synaptic release. The decrease in release probability in $Syt11^{-/-}$ neurons may relate to a reduced number of presynaptic Cav2.2 channels due to impaired trafficking and/or stabilization of these channels.

GBRs inhibit neurotransmitter release by reducing the activity of presynaptic Cav channels (Wu and Saggau, 1997). The GBR agonist baclofen significantly reduced the frequency and amplitudes of sEPSCs in both $Syt11^{+/+}$ and $Syt11^{-/-}$ neurons in culture (Fig. 5D–F). However, baclofen was significantly less efficient in reducing the sEPSC frequencies in $Syt11^{-/-}$ compared to $Syt11^{+/+}$

neurons ($Syt11^{+/+}$: 75.5 ± 3.5% vs $Syt11^{-/-}$: 59.9 ± 5.2%, $p = 0.0155$, Mann–Whitney U test), while the reduction in sEPSC amplitudes was similar ($Syt11^{+/+}$: 27.2 ± 4.5% vs $Syt11^{-/-}$: 36.7 ± 4.0%, $p = 0.1743$, Mann–Whitney U test). Plotting the number of sEPSCs against EPSC amplitudes revealed that in $Syt11^{+/+}$ neuronal cultures, baclofen significantly reduced both large and small amplitude events (Fig. 5G). In contrast, in $Syt11^{-/-}$ cultures, baclofen failed to significantly reduce small amplitude events (Fig. 5H). This indicates a reduction in the number of synapses exhibiting GBR-mediated inhibition of glutamate release in the absence of Syt11. We next tested whether the decreased release probability in $Syt11^{-/-}$ neuronal cultures is due to an increase in the constitutive or tonic activity of GBRs. Application of the inverse agonist CGP54626 did not significantly change the baseline frequency of sEPSCs in either genotype (Fig. EV5A,B), providing no evidence for constitutive or tonic GBR activity in our neuronal cultures. In addition, CGP54626 did not increase the frequency of sEPSCs above baseline when applied after baclofen (Fig. EV5C). In conclusion, the disruption of Syt11-mediated Cav2.2/GBRs complex formation and trafficking reduces both synaptic release probability and GBR-mediated inhibition of glutamate release.

## Hippocampal $Syt11^{-/-}$ neurons exhibit a significant deficit in presynaptic Cav2.2 channels

Various types of Cav channels contribute to activity-dependent neurotransmitter release at brain synapses, with Cav2.1 and Cav2.2 channels being most prominent (Cao and Tsien, 2010; Dunlap et al, 1995; Li et al, 2007; Reid et al, 1997). To determine the contribution of Cav2.2 channels to synaptic transmitter release in $Syt11^{-/-}$ neurons and explore potential compensatory changes in Cav2.1 channels, we measured sEPSC frequencies in the consecutive presence of ω-conotoxin (blocking Cav2.2 channels), ω-conotoxin + ω-agatoxin (blocking Cav2.1 channels), and ω-conotoxin + ω-agatoxin + TTX (preventing action potential-dependent release) (Fig. 6A). Blocking Cav2.2 channels by ω-conotoxin significantly reduced the frequency of sEPSCs and shifted the distribution of inter-event intervals toward larger values in both $Syt11^{-/-}$ and $Syt11^{+/+}$ neuronal cultures (Fig. 6B,C). However, ω-conotoxin was significantly less efficient in inhibiting the sEPSC frequency in $Syt11^{-/-}$ neurons compared to $Syt11^{+/+}$ neurons (Fig. 6D), consistent with a reduction in presynaptic Cav2.2 channels. Notably, the combined inhibitory effect of ω-agatoxin and TTX, but not of ω-agatoxin alone, was significantly larger in $Syt11^{-/-}$ neurons when

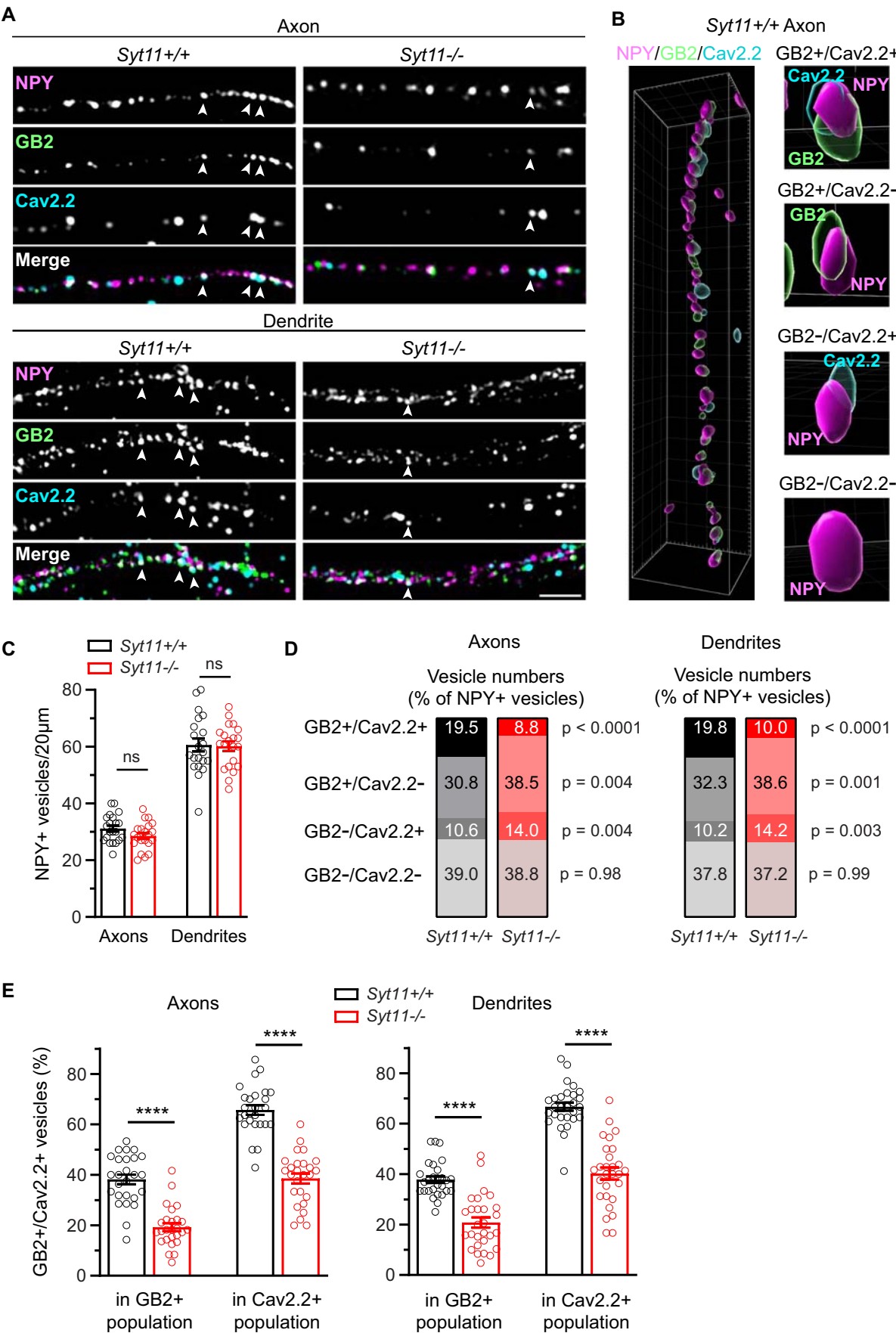

◄ **Figure 4.  Syt11 facilitates recruitment of GBRs and Cav2.2 channels into post-Golgi vesicles.**

(A) Representative single plane SIM images of an axon and dendrite of a cultured hippocampal neuron (DIV14). After fixation and permeabilization, neurons were stained for endogenous NPY (magenta), GB2 (green), and Cav2.2 (cyan). Staining of the α1B subunit of Cav2.2 channels was performed with the anti-Cav2.2 antibody from Alamone Labs (#ACC-002, RRID:AB2039766), which was validated with *CNCNA1b*$^{-/-}$ mouse tissue (Murakami et al, 2007). Arrowheads indicate examples of NPY+ vesicles carrying GB2 and Cav2.2. Scale bar: 5 µm. (B) Representative 3D reconstruction of a stack of SIM images showing the distribution of NPY (magenta), GB2 (green), and Cav2.2 (cyan) in the *Syt11*$^{+/+}$ axon depicted in (A). Examples of four NPY+ vesicle populations are shown on the right: Double positive (GB2+/Cav2.2+), GB2 single positive (GB2+/Cav2.2-), Cav2.2 single positive (GB2-/Cav2.2+), and double negative (GB2-/Cav2.2-). (C) Quantification of the NPY+ vesicle-density in axons and dendrites of *Syt11*$^{+/+}$ and *Syt11*$^{-/-}$ neurons. A total of $n = 21$ neurons per genotype from 4 independent preparations were analyzed. (D) Stacked bar chart illustrating the contribution of the four vesicle populations to NPY+ vesicles. The proportion of double positive (GB2+/Cav2.2+) vesicles is significantly reduced in *Syt11*$^{-/-}$ axons and dendrites, with a concomitant increase in the proportion of GB2 single positive and Cav2.2 single positive vesicles. Numbers are presented as a percentage of the total number of NPY+ vesicles. (E) Analysis of NPY+ vesicle populations. The proportion of double positive (GB2+/Cav2.2+) vesicles is significantly reduced in both the GB2+ and Cav2.2+ populations in the axons and dendrites of *Syt11*$^{-/-}$ neurons. Axons, $n = 26$ neurons; dendrites, $n = 28$ neurons from 4 independent preparations. Data information: Data are presented as mean ± SEM. Statistical significance was determined by unpaired Student's *t*-test (C) or Mann–Whitney U test (D and E). ns not significant; ****$p < 0.0001$. Source data are available online for this figure.

applied after ω-conotoxin (Fig. 6D,E). Altogether, this indicates a reduction in the relative contribution of presynaptic Cav2.2 channels to synaptic release in *Syt11*$^{-/-}$ neurons, partially offset by an upregulation of other Cav channels. In addition, the lack of significant differences in TTX-insensitive mEPSCs between genotypes (Fig. 6D) indicates comparable synapse density.

## Discussion

GPCRs signal to various effectors, including ion channels and second messenger-generating enzymes. GBRs are GPCRs that have evolved to signal through K$^+$ and Ca$^{2+}$ channels, requiring faster activation and deactivation kinetics than second-messenger mediated signaling. GBR-induced inhibition of Cav-type Ca$^{2+}$ channels constitutes a major form of synaptic regulation (Chalifoux and Carter, 2011). GBRs inhibit Cav channels through the activated Gβγ subunits of the G protein, which directly bind to the pore-forming channel subunit (Catterall and Few, 2008). Proteomic studies showed that GBRs physically assemble with their effector Cav2.2 channels into multi-protein signaling complexes (Schwenk et al, 2016). Direct assembly of receptor and effector channel enables faster channel gating than random collision of signaling components (Bhandari et al, 2021; Ciruela et al, 2010; Laviv et al, 2011; Schwenk et al, 2016; Wright et al, 2017). However, it remains unclear how the assembly of this multi-protein complex from low-abundance components can be achieved in the crowded molecular environment of the presynaptic plasma membrane (Bucurenciu et al, 2010; Südhof, 2012). Our findings reveal that Syt11 facilitates the early formation of GBR/Cav2.2 signaling complexes during biogenesis by recruiting GBRs and Cav2.2 channel subunits together into post-Golgi vesicles. This increases the local concentration and proximity of the signaling components in the vesicular membrane, which facilitates assembly of the GBR/Cav2.2 signaling complex (Cebecauer et al, 2010). In the absence of Syt11, both assembly and trafficking of GBR/Cav2.2 complexes are disrupted, resulting in synaptic alterations in neuronal cultures. We observed a decrease in neurotransmitter release probability in *Syt11*$^{-/-}$ neurons. In both, cultured *Syt11*$^{-/-}$ neurons and heterologous cells expressing GBRs and KCTD16 without Syt11, we detected no increase in tonic or constitutive GBR activity, which could have explained the reduced release. However, subtype-selective Cav channel blockers revealed a significant reduction in presynaptic Cav2.2 channels in *Syt11*$^{-/-}$ neurons, which are not fully

compensated for by other Cav channels. This is consistent with the finding that Syt11 exhibits a preference for binding to Cav2.2 channels over Cav2.1 channels (this study), and that presynaptic slots accepting Cav2.2 channels reject Cav2.1 channels (Cao and Tsien, 2010). In addition, baclofen-mediated inhibition of glutamate release was reduced due to a lower number of presynaptic GBRs. These synaptic alternations may contribute to the impaired synaptic plasticity observed in the hippocampus of conditional *Syt11*$^{-/-}$ mice, which specifically lack Syt11 in excitatory forebrain neurons (Shimojo et al, 2019). While our study primarily focused on the presynaptic effects of Syt11 deficiency, it is worth noting that we also observed co-transport of GBRs and Cav2.2 in post-Golgi vesicles within the dendrites. This suggests that the assembly of GBR/Cav2.2 complexes occurs prior to their delivery to the plasma membrane in dendritic regions as well.

As supported by previous proteomic analysis (Schwenk et al, 2016), Syt11 interacts with the auxiliary GBR subunit KCTD16, which serves as a link between GBRs and Cav2.2 channels. KCTD16 is a multi-domain protein that forms homo- and heteropentamers (Fritzius et al, 2017; Zuo et al, 2019) and functions as both a regulatory and scaffold protein (Pin and Bettler, 2016; Schwenk et al, 2016). KCTD proteins associate with GBRs at the ER membrane (Ivankova et al, 2013), where the intracellular β subunit of the Cav channel assembles with the pore-forming α1 subunit (Dolphin and Lee, 2020). Assembly with the β subunit protects the α1 subunit from ER-associated degradation, facilitating the forward trafficking of the channel complex (Dolphin and Lee, 2020; Waithe et al, 2011). Since we observe a reduced co-localization of GBRs and Cav2.2 in post-Golgi vesicles of *Syt11*$^{-/-}$ neurons, we propose that Syt11 recruits Cav2.2 channel subunits together with KCTD16-bound GBRs from the trans-Golgi network into vesicles. Efficient trafficking of Cav channels to release sites requires proteolytic processing of a single precursor protein into a disulfide-bonded α2δ subunit along the biosynthetic pathway (Hoppa et al, 2012; Kadurin et al, 2016; Nieto-Rostro et al, 2018). Notably, α2δ co-purified with the α1B subunit in both anti-GBR and anti-KCTD16 APs (Schwenk et al, 2016). This suggests that all components of GBR/Cav2.2 signaling complexes assemble in Syt11 vesicles. Subsequently, these vesicles process and traffic the signaling complex to presynaptic sites. The available data suggest that Syt11 resides in a specific class of DCVs or cargo vesicles, similar to the related Syt4 (Bajaj et al, 2022; Bharat et al, 2017; Dean et al, 2009; Dean et al, 2012b; Shimojo et al, 2019; Stucchi et al, 2018); and this study). The Syt11 interactome identified in our

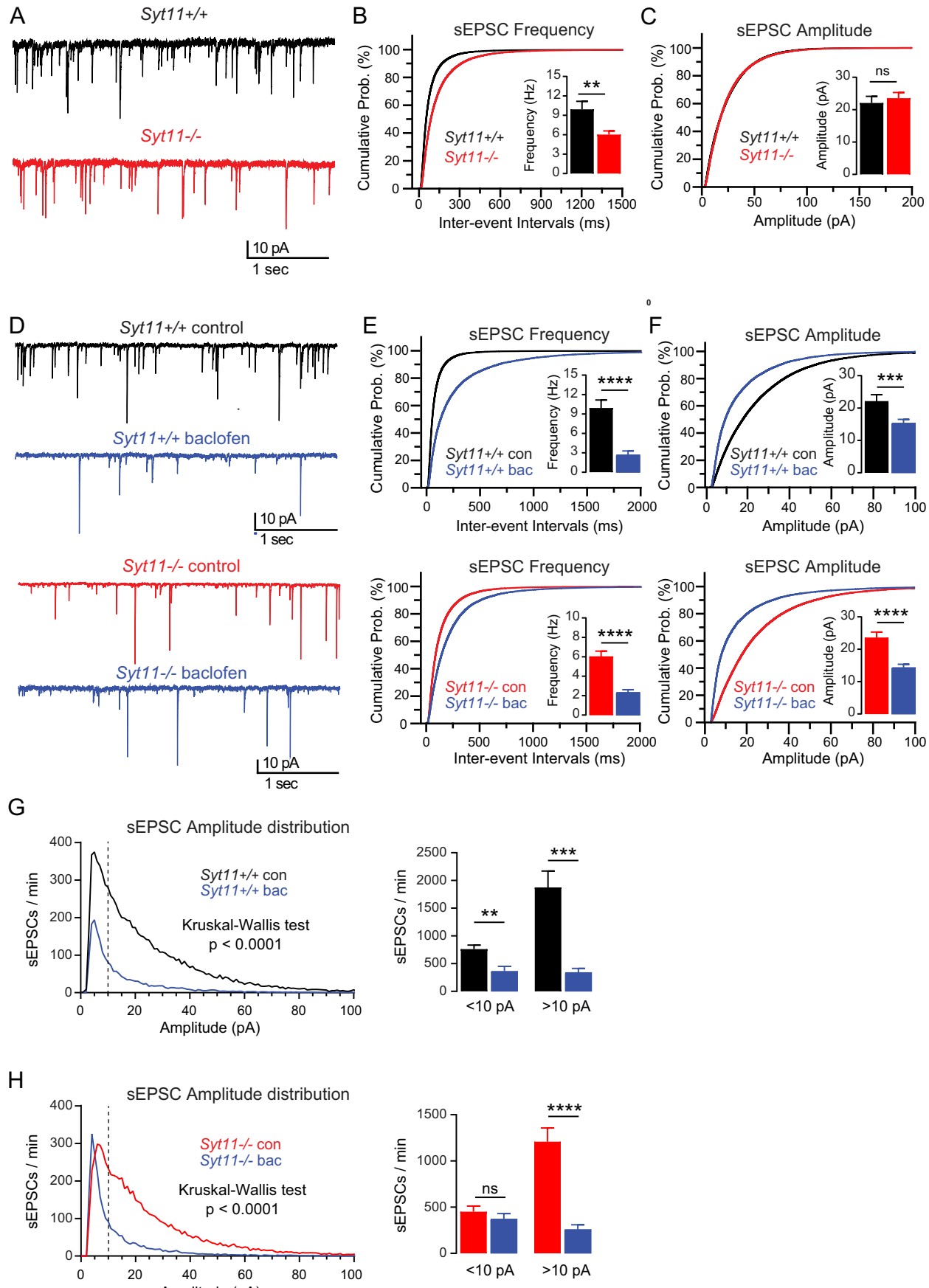

◀ **Figure 5. Reduced synaptic transmission and GBR-mediated inhibition of glutamate release in cultured *Syt11*⁻/⁻ hippocampal neurons.**

(A) Representative traces of sEPSCs from $Syt11^{+/+}$ (black) and $Syt11^{-/-}$ (blue) cultured primary hippocampal neurons recorded in the presence of gabazine (10 μM) at DIV15-19. (B) Cumulative probability distribution of sEPSC inter-event intervals in $Syt11^{+/+}$ and $Syt11^{-/-}$ neurons. The sEPSC frequency (inset) was significantly reduced in $Syt11^{-/-}$ compared to $Syt11^{+/+}$ neurons ($Syt11^{+/+}$: 9.92 ± 1.25 Hz vs $Syt11^{-/-}$: 6.03 ± 0.56 Hz). (C) Cumulative probability distribution of sEPSC amplitudes in $Syt11^{+/+}$ and $Syt11^{-/-}$ neurons. The average sEPSC amplitude (inset) was not significantly different between the genotypes ($Syt11^{+/+}$: 22.06 ± 2.06 pA vs $Syt11^{-/-}$: 23.55 ± 1.73 pA). (D) Representative traces of sEPSCs recorded from a $Syt11^{+/+}$ (top) and $Syt11^{-/-}$ (bottom) neuron in the presence of gabazine (10 μM) before (control, black/red) and after application of baclofen (100 μM, blue). (E) Cumulative probability distributions of sEPSC inter-event intervals from $Syt11^{+/+}$ (top) and $Syt11^{-/-}$ (bottom) neurons. In both genotypes, the sEPSC frequency (insets) was significantly reduced in the presence of baclofen (bac) compared to control (con). Upper inset: $Syt11^{+/+}$ neurons (con: 9.92 ± 1.25 Hz vs bac: 2.66 ± 0.62 Hz). Lower inset: $Syt11^{-/-}$ neurons (con: 6.03 ± 0.56 Hz vs bac: 2.32 ± 0.29 Hz). (F) Cumulative probability distributions of sEPSC amplitudes from $Syt11^{+/+}$ (top) and $Syt11^{-/-}$ (bottom) neurons. In both genotypes, the average amplitude (insets) was significantly decreased in the presence of baclofen (bac) compared to control (con). Upper inset: $Syt11^{+/+}$ neurons (con: 22.06 ± 2.06 pA vs bac: 15.23 ± 1.18 pA). Lower inset: $Syt11^{-/}$ neurons (con: 23.55 ± 1.73 pA vs bac: 14.06 ± 1.10 pA). (G) Number of sEPSCs plotted against sEPSC amplitudes in $Syt11^{+/+}$ neurons in the presence and absence (con) of baclofen (left). Activation of GBRs significantly inhibits small amplitude (<10 pA) and large amplitude (>10 pA) events (right). (H) Number of sEPSCs plotted against sEPSC amplitudes in $Syt11^{-/-}$ neurons in the presence and absence (con) of baclofen (left). Activation of GBRs significantly inhibits large amplitude (>10 pA) but not small amplitude (<10 pA) events. Data information: Data are presented as mean ± SEM. Statistical significance was determined by Mann–Whitney U test (B), unpaired Student's *t*-test (C), paired Student's *t*-test (E, upper inset and F) or Wilcoxon matched-pairs signed-rank test (E, lower inset; G and H). ns not significant; **$p < 0.01$, ***$p < 0.001$, ****$p < 0.0001$. $Syt11^{+/+}$, $n = 14$ neurons; $Syt11^{-/-}$, $n = 19$ neurons from 6 preparations. Source data are available online for this figure.

study comprises a number of transmembrane proteins that are not part of GBR-complexes, such as for example TrkB (Zahavi et al, 2021). This suggests that Syt11 also selects GBR-unrelated cargo for transport in Golgi-derived vesicles. Transport of Syt11 vesicles along microtubules is likely dependent on the kinesin-3 Kif1a (Lo et al, 2011; Stucchi et al, 2018) (and this study) or APP, which links to kinesin-1 motors (Dinamarca et al, 2019; Eggert et al, 2018; Fu and Holzbaur, 2013; Maday et al, 2014). Previous studies provided evidence for the axonal transport of pre-assembled complexes that comprise components of the presynaptic active zone (Ahmari et al, 2000; Dresbach et al, 2006; Shapira et al, 2003). Our study extends this concept to a GPCR/Cav2.2 signaling complex that regulates neurotransmitter release. The Syt11-interactome includes neurexin-1α and calsyntenin-3, both of which are components of trans-synaptic organizing complexes (Liu et al, 2022; Luo et al, 2020; Luo et al, 2021; Pettem et al, 2013; Südhof, 2017). It remains to be determined whether the association of Syt11 with these proteins serves to localize GBR/Cav2.2-complexes at specific synaptic sites.

In summary, our findings indicate that Syt11 orchestrates the assembly of presynaptic GBR/Cav2.2 channel complexes in post-Golgi transport vesicles. Moreover, Syt11 reduces endocytosis of these complexes from the synaptic membrane, consistent with its reported function as an endocytosis clamp (Wang et al, 2016). This coordinated regulation of biogenesis, vesicular transport and surface stability of GBR/Cav2.2 signaling complexes results in an increased release probability, while simultaneously maintaining efficient and robust GBR-mediated inhibitory control over release.

## Methods

### Animals

*Syt11* knockout mice in the C57BL/6J background were generated using CRISPR/Cas9 technology deleting sequence containing the second coding exon between two Cas9 target sequences in the *Syt11* gene (MGI:1859547). Cas9 target sequences were selected using the CRISPOR search algorithm http://crispor.tefor.net. Upstream target sequence: TGC ACA CGG CAG GAG CTG CGA GG (on anti-sense strand); down-stream target sequence: TCC ATG GAT

GAA CTG CCA GGA GG (on anti-sense strand). Deletion of the second coding exon from Val13 to Gln286 (containing the transmembrane and C2A domains) was verified by sequencing. As reported previously, deletion of the second coding exon causes perinatal lethality. Animals were housed on a 12-h light/dark cycle with unrestricted access to food and water. All animal experiments were conducted in compliance with ethical regulations following Swiss guidelines and approved by the veterinary office of Basel-Stadt (reference numbers: 1897_31476 and 1897_35196). The ARRIVE Essential 10 guidelines for reporting animal experimentation were followed in this manuscript.

### Primary cell cultures

Primary hippocampal neurons were prepared from single embryos at embryonic day 16.5. Following dissection hippocampi were digested with 0.25% trypsin (Cat# 15090046, Gibco) in HBSS (Cat# 14170088, Gibco) for 10 min at 37 °C and briefly washed 3 times in Minimum Essential Medium Eagle (Cat# M4655, Sigma) supplemented with 0.6% glucose (Cat# G8769, Sigma) and 10% horse serum (Cat# ECS0091D, Euroclone) to inactivate trypsin. Following trituration, cells were plated at a density of 30,000 cells/cm² on poly-l-lysine hydrobromide-coated glass coverslips in 24-well plates or on μ-Slide 4-well ibiTreat chambered coverslips (Cat# 80426, Ibidi) for live cell imaging. Two hours after plating, the medium was changed to Neurobasal medium (Cat# 21103049, Gibco) supplemented with B27 (Cat# 17504044, Gibco) and 2 mM Glutamax (Cat# 35050038, Gibco). Primary hippocampal neurons were maintained in a humidified incubator with 5% $CO_2$ at 37 °C.

### Cell lines

Human HEK293T cells were directly obtained from ATCC (CVCL_0063) and cultured in a humidified atmosphere of 5% $CO_2$ at 37° in DMEM (Cat# Gibco, 61965) supplemented with 10% FBS (Cat# 10270106, Gibco) and 2% penicillin/streptomycin (Cat# P4333, Sigma).

HEK293T cells stably expressing Gαqi were a gift from the laboratory of Murim Choi (Seoul National University College of Medicine, Republic of Korea) (Yoo et al, 2017). All cell lines were authenticated using Short Tandem Repeat (STR) analysis by

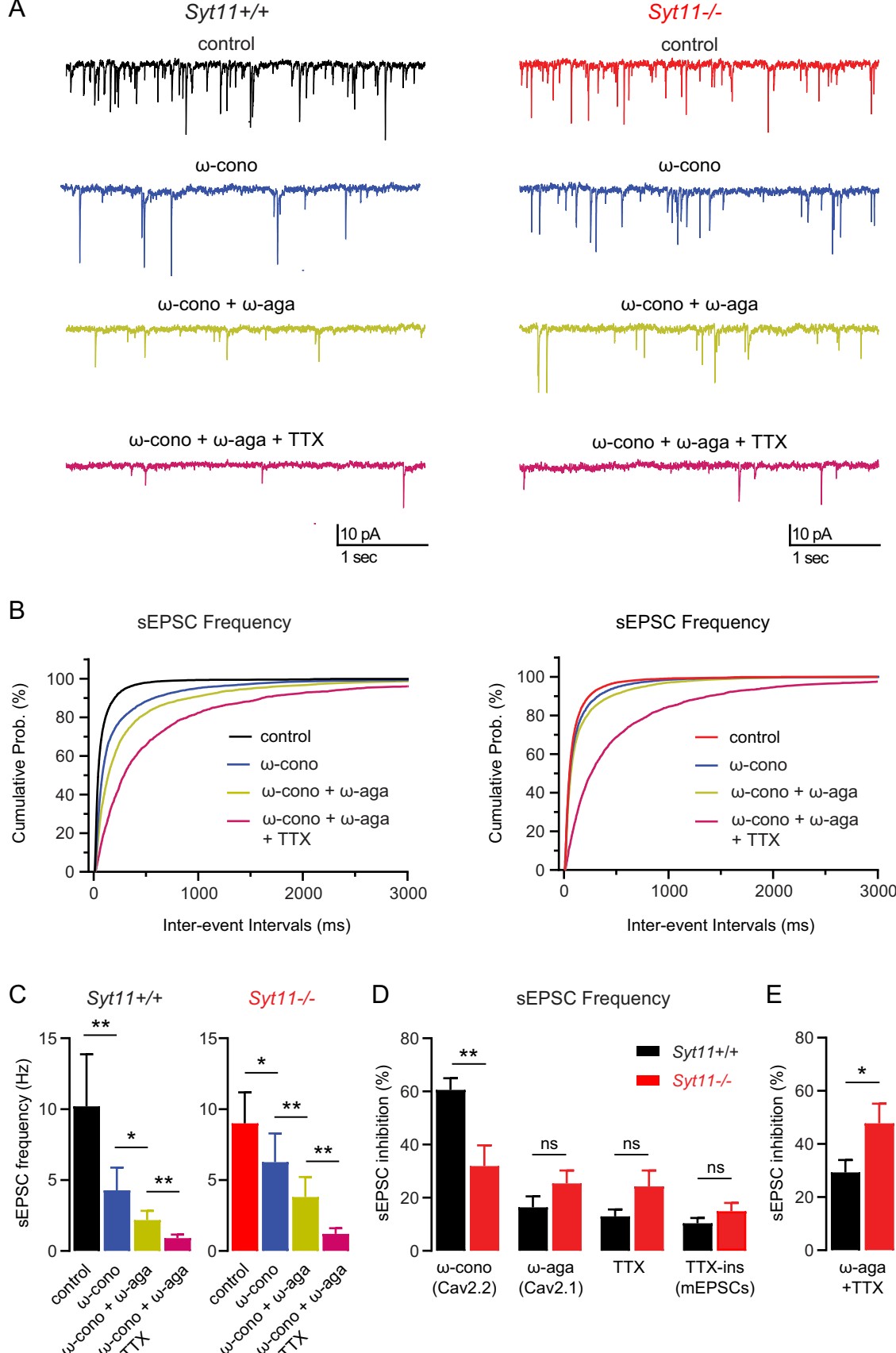

◀

**Figure 6. Cultured *Syt11*−/− hippocampal neurons exhibit a deficit in presynaptic Cav2.2 channels.**

(A) Representative traces of sEPSCs from *Syt11*+/+ and *Syt11*−/− hippocampal neurons in culture recorded at DIV15-19 in the presence of gabazine (10 μM) before (control, black/red) and after application of ω-conotoxin (1 μM, blue), ω-conotoxin + ω-agatoxin (500 nM, yellow) and ω-conotoxin + ω-agatoxin + TTX (1 μM, magenta). (B) Cumulative probability distributions of sEPSC inter-event intervals of *Syt11*+/+ and *Syt11*−/− neurons recorded as in (A). (C) Summary bar graph depicting the sEPSC frequency of *Syt11*+/+ and *Syt11*−/− neurons recorded as in (A). In both genotypes, the frequency of sEPSCs was significantly reduced by the application of ω-conotoxin, ω-conotoxin + ω-agatoxin, and ω-conotoxin + ω-agatoxin + TTX. (D) Summary bar graph depicting the percentage inhibition of sEPSC frequency by ω-conotoxin, ω-agatoxin, and TTX in cultured hippocampal neurons of *Syt11*+/+ (black) and *Syt11*−/− mice (red). Inhibition by ω-conotoxin (blocking Cav2.2 channels) is significantly reduced in *Syt11*−/− compared to *Syt11*+/+ neurons (*Syt11*+/+: 60.55 ± 4.42% vs *Syt11*−/−: 31.90 ± 7.80%). The ω-agatoxin-sensitive (blocking Cav2.1 channels) and TTX-sensitive components of inhibition, as well as the TTX-insensitive component (mEPSCs) show no significant difference between genotypes. (E) The combined ω-agatoxin- and TTX-sensitive component of inhibition is significantly increased in *Syt11*−/− compared to *Syt11*+/+ neurons (*Syt11*+/+: 29.33 ± 4.64% vs *Syt11*−/−: 47.78 ± 7.37%). Data information: Data are presented as mean ± SEM. Statistical significance was determined by Wilcoxon matched-pairs signed-rank test with Bonferroni correction for multiple comparison (C) or Mann–Whitney U test (D and E). ns not significant; *$p < 0.05$, **$p < 0.01$. $n = 9$–11 neurons per genotype from 4 preparations. Source data are available online for this figure.

Microsynth (Switzerland) and tested negative for mycoplasma contamination.

## Affinity purification and mass-spectrometry

Brains from WT mice and Syt11 knock-out mice at the age of P5 were dissected and shock-frozen in liquid nitrogen. Brains were thawed in 10 mM Tris/HCl pH 7.5, 300 mM Sucrose, 1.5 mM MgCl$_2$, 1 mM EGTA, 1 mM Iodoacetamide and protease inhibitors (Aprotinin, Leupeptin, Pepstatin A, PMSF) and homogenized with 15 ml Dounce homogenizer. Tissue lysates were centrifuged (4 min, 1000 × $g$) and respective supernatants ultracentrifuged (20 min, 200,000 × $g$). Pellets were homogenized in lysis buffer (5 mM Tris/HCl pH 7.4, 2 mM Iodoacetamide) and again subjected to ultracentrifugation (20 min, 200,000 × $g$). Membrane pellets were resuspended in 10 mM Tris/HCl pH 7.4, 0.5 M Sucrose, filled in thin layer UC-tubes and thoroughly underlayed with 10 mM Tris/HCl pH 7.4, 0.5 M sucrose buffer and 10 mM Tris/HCl pH 7,4, 1.3 M Sucrose buffer and ultracentrifuged (45 min, 30,000 rpm). Membrane proteins were harvested at the interface (0.5/1.3 M sucrose), washed and resuspended in 20 mM Tris/HCl pH 7.4. Protein concentrations were measured by Bradford assay. For each affinity purification (AP) 1 mg membranes were solubilized in 1 ml CL-91 (Cat# CL-91–01, Logopharm) supplemented with 2 mM Ca$^{2+}$ and protease inhibitors (see above). Insoluble proteins were removed by ultracentrifugation (10 min, 125,000 × $g$) and solubilisates were incubated for 2 h with 10 μg antibodies pre-coupled to protein A Dynabeads (Cat# 10002D, Invitrogen). The following antibodies were used: rabbit anti-Syt11 (Cat# 270 003; Synaptic Systems), rabbit IgG (Cat# 12-370, Millipore). Anti-Syt11 antibodies were incubated with WT solubilisate. As control served incubations of anti-Syt11 in Syt11 knock-out solubilisate and IgGs in WT solubilisate. Unbound proteins were removed by two brief washing steps with 2× 0.5 ml CL-91. Bound proteins were eluted with 10 μl Laemmli buffer w/o DTT and subsequently, shortly separated on SDS-PAGE. Gels were silverstained to visualize proteins and enable accurate cutting of the gel lanes in two pieces (low and high molecular weight). Experiments were done in duplicates.

Then, proteins were in-gel digested with sequencing grade modified trypsin (Cat# V5111, Promega). Peptides were extracted, vacuum-dried and dissolved in 13 μL of 0.5% (v/v) trifluoroacetic acid. The analyses of tryptic peptide mixtures were carried out on an Orbitrap Elite mass spectrometer coupled to an UltiMate 3000 RSLCnano HPLC system (Thermo Scientific) as described

(Kocylowski et al, 2022). Appropriate amounts were loaded and LC gradients were built with eluent 'A' (0.5% (v/v) acetic acid in water) and eluent 'B' (0.5% (v/v) acetic acid in 80% (v/v) acetonitrile/20% (v/v) water): 5 min 3% 'B', 60 min from 3% 'B' to 30% 'B', 15 min from 30% 'B' to 99% 'B', 5 min 99% 'B', 5 min from 99% 'B' to 3% 'B', 15 min 3% 'B'. Eluting peptides were electrosprayed at 2.3 kV (positive polarity) via Nanospray Flex ion sources into (CID fragmentation of the 10 most abundant at least doubly charged new precursors per scan cycle) and analyzed with the following major settings: MS2 injection time 200 ms, intensity threshold for fragmentation were 2000 counts. LC-MS/MS RAW files were converted into peak lists (Mascot generic format, mgf) with ProteoWizard msConvert (https://proteowizard.sourceforge.io/) and searched with Mascot Server 2.6.2 (Matrix Science Ltd, London, UK) against a database containing all mouse, rat, and human entries of the UniProtKB/Swiss-Prot database. Initially broad mass tolerances were used. Based on the search results peak lists were linear shift mass recalibrated using in-house developed software and searched again with narrow mass tolerances for high-resolution peaks (peptide mass tolerance ±5 ppm; fragment mass tolerance 0.8 Da). One missed trypsin cleavage and common variable modifications were accepted. Default significance threshold ($p < 0.05$) and an expect value cut-off of 0.5 were used for displaying search results.

Protein quantifications were determined according to a label-free procedure (Kocylowski et al, 2022). Briefly, peptide signal intensities (peak volumes, PVs) were extracted from FT full scans and mass calibrated using MaxQuant v1.6.3 (http://www.maxquant.org). The resulting PV table with all protein-specific peptide signal intensities from all runs built the basis for the evaluations. The molecular abundance of proteins identified in eluates of affinity purifications were calculated as abundancenorm-spec values (Bildl et al, 2012). The specificity of protein co-purifications were determined on the basis of target-normalized abundance ratio (tnR) of proteins determined in anti-Syt11 APs from WT versus control APs (see above). tnR values of all proteins consistently identified in two Syt11 APs and reliably detected with an abundancenormspec value above 4000 were plotted in Fig. 1a. Ribosomal proteins were removed. 28 proteins were enriched (tnR-value > 0.18) in both Syt11 APs.

## Molecular biology

FLAG-GB1b, HA-GB2, HA-GB2Y902A and FLAG- and Myc-tagged KCTD plasmids were as described (Dinamarca et al, 2019;

Fritzius et al, 2017; Seddik et al, 2012). To construct Myc-tagged GB2 intracellular C-terminal domain (GB2ICD), amino acids $I_{744}$ to $L_{940}$ of mouse GB2 were cloned in frame with a N-terminal 3xMyc-tag into pCI (Cat# E1731, Promega). To construct Syt11C2-eGFP, the cytoplasmic C2A and C2B domains of Syt11 (Cat# MR206864, OriGene) were cloned into pEGFP-N1 (Cat# 6085-1, Clontech). In the split Venus constructs Syt11-VN, Syt11ΔC2-VN, Syt1-VN and GB2-VC the N-terminal 1-172 (VN) or C-terminal 155-238 (VC) residues of the yellow fluorescent protein Venus (Nagai et al, 2002) were cloned in frame at the C-terminus of the respective proteins separated by the linker sequence PRARDPP-VAT (Armando et al, 2014; Dinamarca et al, 2019). The Syt1 plasmid was from OriGene (Cat# MR206688), pcDNA3.1-mCherry from Addgene (Cat# 128744), NPY-mCherry from Addgene (Cat# 67156) and pSI-AAR6-Rab5a-mcherry was a gift from M. Spiess (Kalin et al, 2015). The Cav2.2 α1B subunit (rat *Cacna1b*) was from Addgene (Cat# 26567), the β3 subunit (human *Cacnb3*) from OriGene (Cat# RC207229) and the α2δ1 subunit (rat *Cacna2d1*) from Addgene (Cat# 26575).

HEK293T cells were transfected at 80–90% confluence using polyethyleneimine (PEI) transfection reagent (Cat# 408727, Sigma) with 2 µg/µl PEI per µg of plasmid DNA. The total amount of DNA in the transfections was kept constant by supplementing with empty pCI plasmid (Cat# E1731, Promega). Cells were harvested 48 h after transfection for co-immunoprecipitation and Western blot analysis. For BiFC experiments HEK293T cells were transiently transfected in Opti-MEM (Cat# 31985047, Gibco) using Lipofecta-mine 3000 (Cat# L3000001, Invitrogen) according to the manu-facturer's instruction. Transfection of primary hippocampal neurons was performed using Lipofectamine 3000. For each well 1 µg of plasmid DNA and 0.4 µl Lipofectamine were added to separate tubes containing 100 µl Neurobasal medium. After 5 min incubation at RT, the two solutions were mixed together, incubated for another 20 min and finally added to the cultures from which all but 200 µl of the conditioned medium has been removed. Following incubation for 45 min at 37 °C, the Lipofectamine/DNA mixture was replaced with the conditioned medium that was kept in the incubator at 37 °C. Neurons were used for live-cell imaging 8 h after transfection or fixed for immunofluorescence analysis 24 h after transfection.

### Affinity purification and western blot analysis

HEK293 cells were harvested 48 h after transfection, washed in ice-cold PBS, and subsequently lysed in a Nonidet P-40 buffer (100 mm NaCl, 1 mm EDTA, 0.5% Nonidet P-40 Substitute (Cat# Sigma, 74385), 20 mm Tris/HCl, pH 7.4) supplemented with complete EDTA-free protease inhibitor mixture (Cat# 11873580001, Roche). After rotation for 10 min at 4 °C, the lysates were cleared by centrifugation at $16,000 \times g$ for 10 min at 4 °C. Cleared lysates were then either directly used for Western blot analysis or incubated for 3 h at 4 °C with 1 µg rabbit anti-Cav2.2 (Cat# ACC-002, Alomone), rabbit anti-Syt11 (Cat# 270 003 Synaptic Systems) or mouse anti-Myc (Cat# 9E10, Santa Cruz) antibodies for immunoprecipitation. To capture antibody-protein complexes lysates were incubated with 1 µl of magnetic Protein G Dynabeads (Cat# 10004D, Invitrogen) for 15 min. Following 5 wash-steps in Nonidet P-40 buffer immunoprecipitated proteins were resolved together with the input lysates using standard SDS-PAGE for 45 min at 70 mV, followed by

additional 2 h at 120 mV. Proteins were transferred using wet transfer to 0.45 µm polyvinylidene fluoride membranes (Cat# IPVH00010, Millipore) for 120 min at 200 mA, blocked for 90 min in PBS, containing 0.1% Tween-20, and 5% skim milk and incubated with antibodies in storage solution (5% BSA in 1x PBS, 0.05% sodium azide). Primary antibodies were mouse anti-Flag (Cat# F1804, Sigma, 1:1000), mouse anti-GFP (Cat# 11814460001, Roche, 1:1000), mouse anti-Myc (Cat# 9E10, Santa Cruz, 1:1000), mouse anti-GABA$_{B1}$ (Cat# ab55051, Abcam, 1:1000), Mouse ant-HA (Cat# MMS-101P-200, Covance, 1:1000) rabbit anti-Cav2.2 (Cat# AB5154, Millipore, 1:1000), rabbit anti-FLAG (Cat# F7425, Sigma), rabbit anti-Myc (Cat# C3956, Sigma) and rabbit anti-GAPDH (Cat# ABS16, Millipore, 1:1000). Secondary antibodies were peroxidase-coupled sheep anti-mouse (Cat# NA931 GE Healthcare, 1:10,000) and AffiniPure mouse anti-rabbit IgG light chain (Cat# 211-032-171, Jackson ImmunoResearch Labs, 1:10,000). Membranes were washed 3× for 10 min at RT with PBS after each antibody incubation. The chemiluminescent substrate SuperSignal West Pico PLUS (Cat# 34580, Thermo Scientific) was used for visualization on a FUSION FX7 EDGE Imaging System (Witec AG).

### Gα and Gβγ signaling assays

To monitor Gα signaling by GBRs, HEK293T cells stably expressing Gαqi were transiently transfected with Flag-GB1b, Flag-GB2, KCTD16, and SRE-FLuc with or without Syt11. Transfected cells were distributed into 96-well microplates (Greiner Bio-One) at a density of 100,000 cells/well. After 18 h, the culture medium was replaced with Opti-MEM™-GlutaMAX™. GB1b/2 receptors were activated with various concentrations of GABA for 6 h. FLuc activity in lysed cells was measured using the Luciferase® Assay Kit (Promega) using a Spark® microplate reader. Lumines-cence signals were adjusted by subtracting the luminescence obtained when expressing SRE-FLuc fusion proteins alone.

To monitor Gβγ signaling by GBRs, HEK293T cells were transiently transfected with Flag-GB1b, Flag-GB2, KCTD16, GRK3ct-RLuc (Hollins et al, 2009; Masuho et al, 2015) Gαo, Gβ2, and Venus-Gγ$_2$ plasmids, with or without Syt11. Transfected cells were seeded into 96-well microplates (Greiner Bio-One) at a density of 100,000 cells/well. After 18 h, cells were washed, and coelenterazine h (5 µM, NanoLight Technologies, Prolume Ltd., Pinetop-Lakeside, United States of America) added for 5 min. Luminescence and fluorescence signals were alternately recorded for a total duration of 640 sec using a Spark® microplate reader. GABA and CGP54626 were injected at 127 and 384 s, respectively, using the Spark® microplate reader injection system. The BRET ratio was calculated as the ratio of the light emitted by Venus-Gγ$_2$ (530–570 nm) over the light emitted by GRK3ct-RLuc (370–470 nm). BRET ratios were adjusted by subtracting the ratios obtained when RLuc fusion proteins were expressed alone. Each data point represents a technical triplicate.

### Immunofluorescence

Primary hippocampal neurons on glass coverslips were fixed at DIV10 or DIV14 for 15 min in 4% PFA/4% sucrose in PBS supplemented with 1 mM MgCl$_2$ and 0.1 mM CaCl$_2$ (PBS$^{+/+}$, Cat# D8662, Sigma), washed in PBS$^{+/+}$, permeabilized and blocked for

1 h at RT in PBS$^{+/+}$ containing 0.2% Triton X-100 (Cat# X100, Sigma) and 5% horse serum and labeled in blocking solution with primary antibodies overnight at 4 °C and secondary antibodies for 1 h at RT. Several washes in PBS$^{+/+}$ were performed after each antibody incubation. Coverslips were mounted on glass slides using ProLong Diamond Antifade Mouuntant (Invitrogen). Primary antibodies were rabbit anti-synaptophysin (Cat# ab32127, Abcam, 1:1000), mouse anti-PSD95 (Cat# MA1-045, Thermo Scientific, 1:1000), chicken anti-MAP2 (Cat# ab5392, Abcam, 1:3000), sheep anti-NPY (Cat# ab6173, Abcam, 1:500), guinea pig anti-GABA$_{B2}$ (Cat# 322 205, Synaptic Systems, 1:1000), rabbit anti-Cav2.2 (Cat# ACC-002, Alomone Labs, 1:1000), rabbit anti-A1R (Cat# ab3460, Abcam, 1:1000) and anti-LAMP1 (Cat# ab208943, Abcam, 1:1000). All Alexa Fluor conjugated species-specific secondary antibodies (Abcam and Thermo Scientific) were used at a dilution of 1:1000.

For transferrin-uptake experiments, neurons at DIV14 were cultured in fresh Neurobasal medium for 30 min and subsequently incubated for 1 h with Alexa Fluor 647-conjugated transferrin (Cat# T23366, Thermo Scientific) at a final concentration of 50 μg/μl. Cultures were maintained in a humidified incubator with 5% CO$_2$ at 37 °C during all incubations and finally processed for immunofluorescence analysis as described above.

## Imaging and analysis

Confocal and live-cell imaging was performed with a Leica point scanning confocal Sp5-II-matrix microscope, using a 63× or 40×/1.40-0.60 PlanApo Lamda Blue objective. Fluorescence filter sets were selected according to the fluorophores used. During live-cell imaging at a rate of 1 frame/s, the temperature and CO$_2$ was controlled. Super-resolution structured illumination (3D-SIM) imaging was with an Applied Precision OMX BLAZE microscope, using a 60×/NA 1.42 PlanApo N objective.

Images were taken under identical acquisition parameters for all conditions within the experiment. Saturation was avoided by using image acquisition software to monitor intensity values. All confocal images were processed by deconvolution using the Huygens Essential software (Scientific Volume Imaging B.V, Netherlands) and analyzed using Fiji or Imaris analysis software. Quantification of co-localization of two fluorophores was performed using the JACoP plug-in of Fiji. The Mander's coefficients were used to express the fraction of intensity in a channel that is located in pixels where there is above zero intensity in the other channel. The Pearson correlation coefficient was used to compare the degree of co-localization of two fluorophores between genotypes. All the analyzed images were taken at the same condition and in order to be unbiased, the threshold was set automatically. Only single plane images were analyzed.

Kymographs for analysis of vesicle transport were created by drawing one-pixel-wide lines traced from the soma to the axon tip or on dendrites using the KimographBuilder plug-in of Fiji. The trafficking velocities were obtained using the Velocity measurement tool. Episodes of directed vesicle movement are represented in kymographs as displacements in the anterograde or retrograde direction. Non-mobile episodes produce straight vertical lines with short horizontal displacements resulting from the "wiggling" of vesicles.

For quantification of Alexa Fluor 647-conjugated transferrin uptake the mean fluorescence/area was determined on images of

isolated neurons excluding the soma due to oversaturation of the signal.

For quantification of super-resolution images, 3D reconstruction of image stacks was performed using the automated surface reconstruction plug-in of Imaris. NPY+ vesicles were then manually assigned to one of the following four groups: (1) Double positive (GB2+, Cav2.2+), (2) GB2 single positive (GB2+, Cav2.2-), (3) Cav2.2 single positive (GB2-, Cav2.2+), and (4) double negative (GB2-, Cav2.2-).

## Electrophysiology

For whole-cell voltage-clamp recordings cell culture coverslips were transferred to a bath chamber and continuously perfused with oxygenated artificial cerebrospinal fluid (ACSF) containing (in mM): 121 NaCl, 25 NaHCO$_3$, 25 glucose, 2.5 KCl, 1.25 NaH$_2$PO$_4$, 1 CaCl$_2$, 4 MgCl$_2$. The ACSF was equilibrated with carbogen (95% O$_2$/5% CO$_2$) at RT (21–24 °C), resulting in pH 7.4. Cells were visualized using an AxioExaminer.D1 (Zeiss) and infrared differential interference contrast video microscopy. Patch pipettes were pulled from borosilicate glass tubing with a 2.0 mm outer diameter and 0.5 mm wall thickness (Hilgenberg) using a Flaming-Brown P-97 puller (Sutter Instruments). Patch pipettes had a resistance between 3.5 and 6.5 MΩ and were filled with an internal solution containing the following (in mM) for recordings of spontaneous postsynaptic currents (sPSCs): 136 Cs-Gluconate, 4 CsCl 10 EGTA, 10 HEPES, 2 MgCl$_2$, 2 Na2ATP, 0.3 GTP, 1 phosphocreatine and 5 mM QX-314.The pH was adjusted to 7.3 with CsOH.

Recordings were obtained using a Multiclamp 700B amplifier (Molecular Devices), filtered at 10 kHz for seal, series and input resistance measurement or 1 kHz for sPSCs recordings, and digitized at 20 kHz with a CED Power 1401 interface (Cambridge Electronic Design). Data acquisition was controlled using IGOR Pro 6.31 (Wave Metrics) and the CFS library support from Cambridge Electronic Design. Recordings were only included if the initial seal resistance was >5 times higher than the input resistance (Rinput) of the cells typically ranging from 2–9 GΩ. Series resistance (Rs = 7–25 MΩ) was not compensated and experiments were discarded if Rs changed by >20% between different drug applications.

Spontaneous glutamatergic PSCs were recorded in the presence of the GABA$_A$ receptor blocker gabazine (SR 95531 hydrobromide, 10 μM) (Cat# 1262, Tocris; 10 mM stock dissolved in water). (RS)-Baclofen (100 μM) (Cat# 0417, Tocris; 100 mM stock dissolved in 1eq NaOH) was used to activate GABA$_B$ receptors. The inverse agonist CGP54626 (4 μM) (Cat# 1088, Tocris; 10 mM stock dissolved in DMSO) was used to block GABA$_B$ receptors. ω-Conotoxin GVIA (1 μM) (Cat# 1085, Tocris; 0.5 mM stock dissolved in 1% PBS) and ω-agatoxin TK (500 nM) (Cat# 2802, Tocris; 0.25 mM stock dissolved in 1% PBS) were used to block Cav2.2 or Cav2.1, respectively. For the dilution of ω-conotoxin GVIA and ω-agatoxin TK, the ACSF was supplemented with 0.1 mg/ml BSA (Cat# A7906-506, Sigma). TTX (1 μM) (Cat# T-550, Alomone Labs; 1 mM stock dissolved in water) was used to block action potentials. All drugs were stored in aliquots at −20 °C and diluted in ACSF prior to the recording.

Patch-clamp data was analyzed offline using the open source analysis software Stimfit (https://neurodroid.github.io/stimfit; Guzman et al, 2014) and customized scripts written in Python. For the

analysis of spontaneous glutamatergic PSCs a template-matching algorithm, implemented in Stimfit (Clements and Bekkers, 1997; Jonas et al, 1993), was used as described previously (Schmidt-Salzmann et al, 2014). Automatically detected events were visually controlled and false positive events were deleted. The remaining events were fitted with the sum of two exponential functions revealing the amplitude, rise time and decay time of the spontaneous PSCs. Rs was determined by giving a −5-mV voltage step for 400 ms in voltage-clamp mode (command potential set at −70 mV) and was monitored throughout the experiments. Rs was calculated by dividing the −5-mV voltage step by the peak current value generated immediately after the step in the command potential. Rinput was calculated by giving a −5-mV step in voltage-clamp mode (command potential set at −70 mV), which resulted in transient current responses. The difference between baseline and steady-state hyperpolarized current (ΔI) was used to calculate Rinput using the following formula: Rinput = −5 mV/ΔI.

## Statistical analysis

Statistical analysis was carried out using GraphPad Prism version 7-9 (GraphPad, La Jolla, CA) and SPSS version 22 (IBM). No statistical methods were used to pre-determine sample sizes. No data was excluded from the analysis unless indicated in the Methods or Quantification details. Group assignment was defined by genotype; thus, no randomization was necessary. No statistical method was used to predetermine sample sizes. During the acquisition and analysis of datasets for quantification, investigators were blinded to genotype. Individual datasets were tested for normality with the D'Agostino-Pearson or Shapiro–Wilk test. For data that passed the normality test, statistical significance was assessed by unpaired or paired two-tailed $t$-test or ANOVA as indicated. Otherwise, we used the Mann–Whitney U or Wilcoxon signed-rank tests for non-normally distributed data. In all tests, probability values of $p < 0.05$ were considered statistically significant. Significance levels are denoted as *$p < 0.05$, **$p < 0.01$, ***$p < 0.001$, ****$p < 0.0001$. The details of statistical tests are described in the Figure legends. Group data are presented as mean ± SEM.

## Data availability

The mass spectrometry proteomics data have been deposited to the ProteomeXchange Consortium via the PRIDE (Perez-Riverol et al, 2022) partner repository with the dataset identifier PXD044764 (https://www.ebi.ac.uk/pride/archive/projects/PXD044764).

The source data of this paper are collected in the following database record: biostudies:S-SCDT-10_1038-S44319-024-00147-0.

## Peer review information

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

## Acknowledgements

We thank Y Tan for contributions at an early stage of the study and P Scheiffele for helpful discussions. We thank NA Lambert, BS Muntean, and KA Martemyanov for providing the assay to monitor Gβγ signaling. This work was supported by grants from the Swiss National Science Foundation (SNF) to BB (31003A-152970, 310030B_201291) and JB (310030-205198), and the German Research Foundation (DFG) to BF (TRR 152—project ID 239283807, FA 332/15-1—project ID 439189341 and Excellence Strategy, CIBSS – EXC-2189—project ID 390939984).

## Author contributions

**Luca Trovò**: Conceptualization; Formal analysis; Investigation; Visualization; Writing—original draft. **Stylianos Kouvaros**: Formal analysis; Investigation; Visualization. **Jochen Schwenk**: Conceptualization; Formal analysis; Investigation; Visualization. **Diego Fernandez-Fernandez**: Formal analysis; Investigation; Visualization. **Thorsten Fritzius**: Formal analysis; Investigation; Visualization. **Pascal Dominic Rem**: Formal analysis; Investigation; Visualization. **Simon Früh**: Formal analysis; Investigation; Visualization. **Martin Gassmann**: Conceptualization; Formal analysis; Investigation; Visualization; Writing—original draft; Writing—review and editing. **Bernd Fakler**: Conceptualization; Funding acquisition; Writing—review and editing. **Josef Bischofberger**: Conceptualization; Formal analysis; Funding acquisition; Writing—review and editing. **Bernhard Bettler**: Conceptualization; Funding acquisition; Writing—original draft; Writing—review and editing.

Source data underlying figure panels in this paper may have individual authorship assigned. Where available, figure panel/source data authorship is listed in the following database record: biostudies:S-SCDT-10_1038-S44319-024-00147-0.

## Disclosure and competing interests statement

BB is a member of the scientific advisory board of Addex Therapeutics, Geneva, a pharmaceutical company focused on the development of allosteric modulators for neurological disorders. All other authors declare no competing interests.

# Expanded View Figures

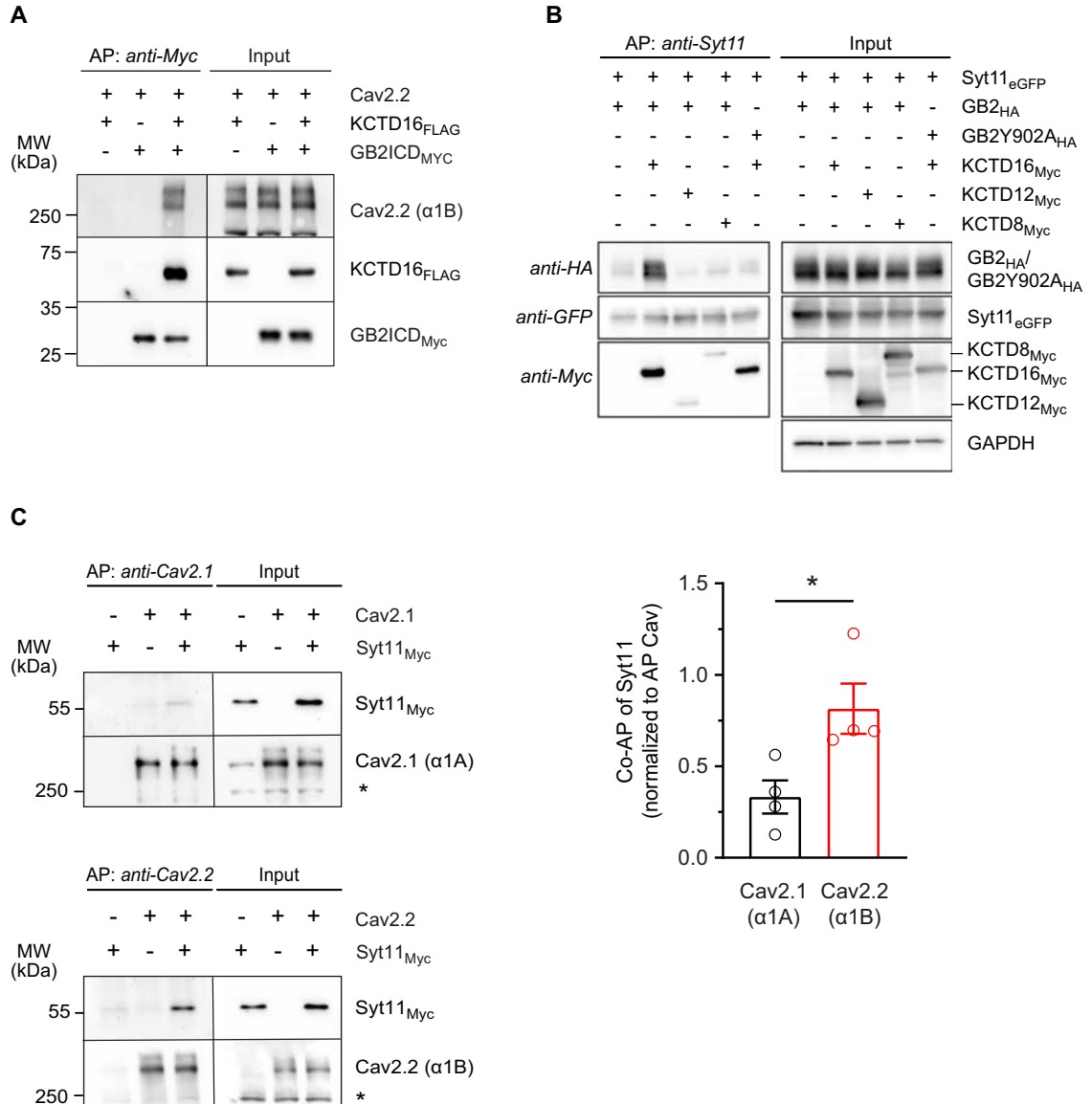

**Figure EV1. Mapping of protein-protein interactions within Syt11/GBR/Cav2.2 complexes in HEK293T cells.**

(A) Cav2.2 co-purifies with the Myc-tagged intracellular C-terminal domain of GB2 (GB2ICD) in the presence of FLAG-tagged KCTD16 from total cell lysates of transfected HEK293T cells. The α1B subunit of Cav2.2 channels was co-expressed with auxiliary β and α2δ subunits. (B) HA-tagged GB2 co-purifies with eGFP-tagged Syt11 in the presence of Myc-tagged KCTD16, but not in the presence of KCTD8 or KCTD12 from total cell lysates of transfected HEK293T cells. HA-tagged GB2Y902A, a GB2 mutant that cannot bind KCTD proteins (Schwenk et al, 2010), does not co-purify with eGFP-tagged Syt11 in the presence of KCTD16. (C) Significantly increased co-purification of Myc-tagged Syt11 with the α1B subunit of Cav2.2 channels compared to α1A subunit of Cav2.1 channels from total cell lysates of transfected HEK293T cells. Auxiliary β and α2δ subunits were co-expressed with the α1A and α1B subunits. Representative Western blots (left) and quantification from $n = 4$ independent experiments (right). Values are presented as mean ± SEM, *$p = 0.028$, Mann–Whitney U test.

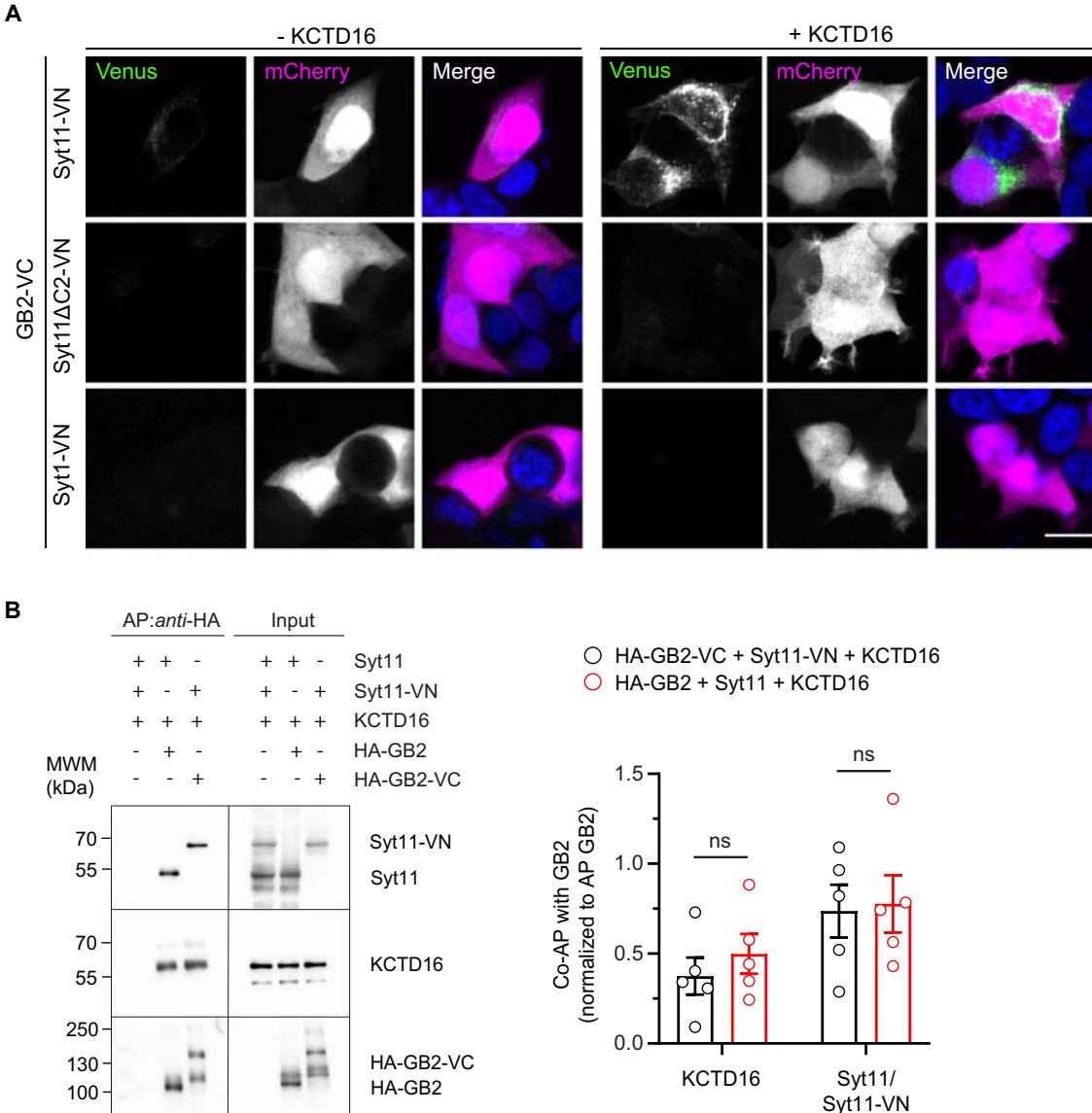

**Figure EV2. Validation of the GB2-VC/Syt11-VN BiFC in transfected HEK293T cells.**

(A) Representative confocal images of HEK293T cells expressing GB2-VC and Syt11-VN tagged with the C-terminal (VC) and N-terminal (VN) fragments of the fluorescent Venus protein (top row). Reconstitution of Venus fluorescence is observed only in cells expressing KCTD16. In control experiments, replacing Syt11-VN with Syt11ΔC2-VN lacking the C2A and C2B domains (middle row) or Syt1-VN (bottom row) does not reconstitute Venus fluorescence. Transfected cells were identified using mCherry. Scale bar: 10 μm. (B) Representative Western blots (left) and corresponding quantifications from $n = 5$ independent experiments (right) of APs with anti-HA antibodies from cell lysates of transfected HEK293T cells expressing the indicated constructs. AP and input lanes were probed with anti-Syt11 (top), anti-KCTD16 (middle), and anti-HA (bottom) antibodies. The presence of VN- or VC-tags on Syt11 and GB2, respectively, does not significantly alter the amounts of KCTD16 ($p = 0.436$) and Syt11 ($p = 0.858$) co-purified with GB2. Values are presented as mean ± SEM, ns = not significant, unpaired t-test.

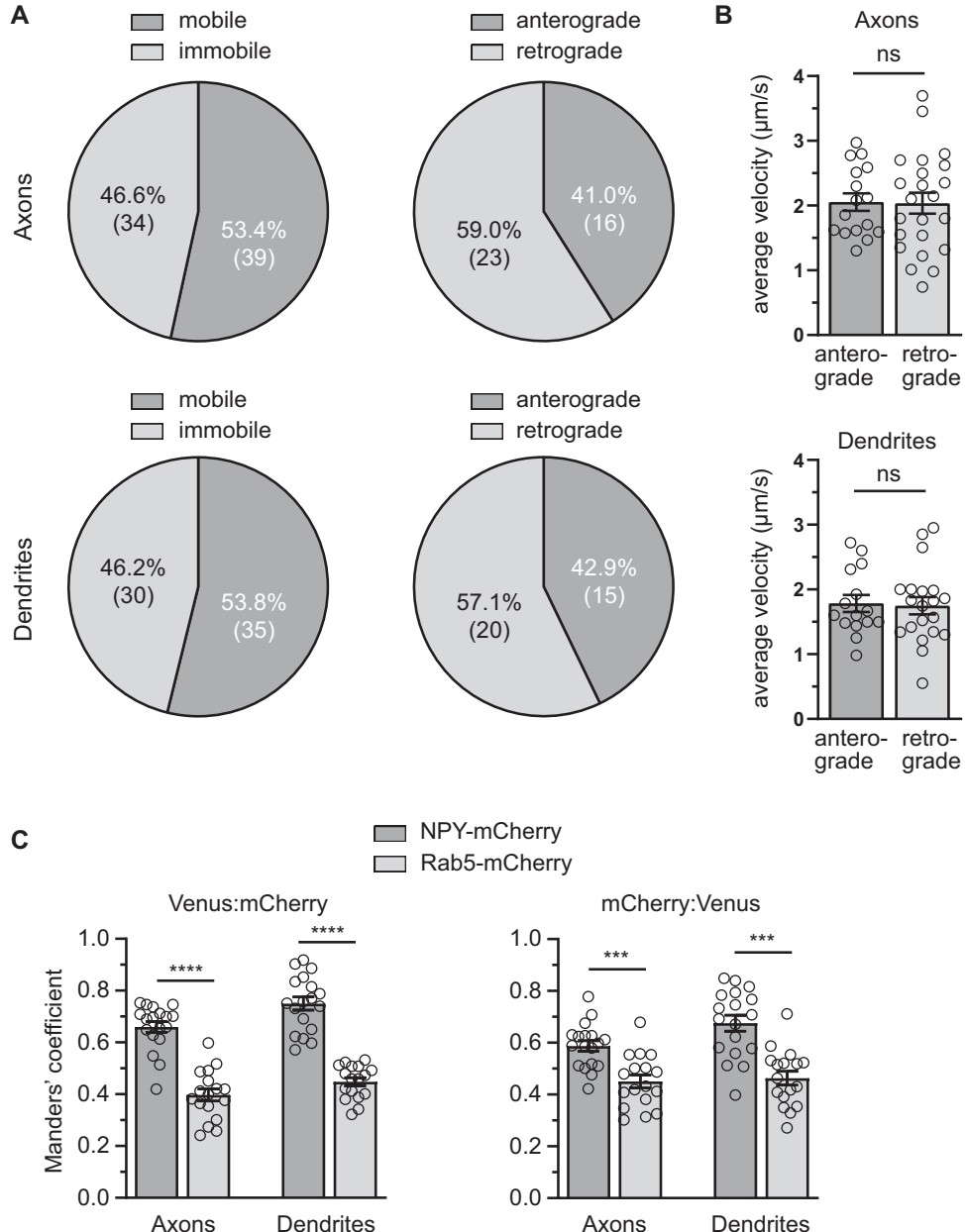

**Figure EV3.    Trafficking analysis of GB2-VC/Syt11-VN complexes in axons and dendrites of cultured hippocampal neurons.**

(**A**) Live-cell imaging analysis of GB2-VC/Syt11-VN complexes (Venus fluorescence) in transfected neurons. Left: Percentage of mobile and immobile complexes. Right: Percentage of complexes traveling antero- and retrograde. The number of complexes analyzed is indicated in brackets. Data are from 4 independent transfections. (**B**) Average velocities of GB2-VC/Syt11-VN complexes traveling antero- and retrograde. Axons: anterograde, $n = 16$ complexes; retrograde, $n = 23$. Dendrites: anterograde, $n = 15$; retrograde, $n = 20$. (**C**) Co-localization of GB2-VC/Syt11-VN complexes and mCherry-tagged NPY or Rab5 in transfected neurons. The Manders' coefficients report the degree of overlap between Venus and mCherry fluorescence. NPY-Cherry, $n = 18$ neurons; Rab5-mCherry, $n = 17$ neurons from 3 independent transfections. Data information: Data are presented as mean ± SEM. Statistical significance was determined by two-way ANOVA (**B**) or Mann–Whitney U test (**C**). ns not significant; ***$p < 0.001$, ****$p < 0.0001$.

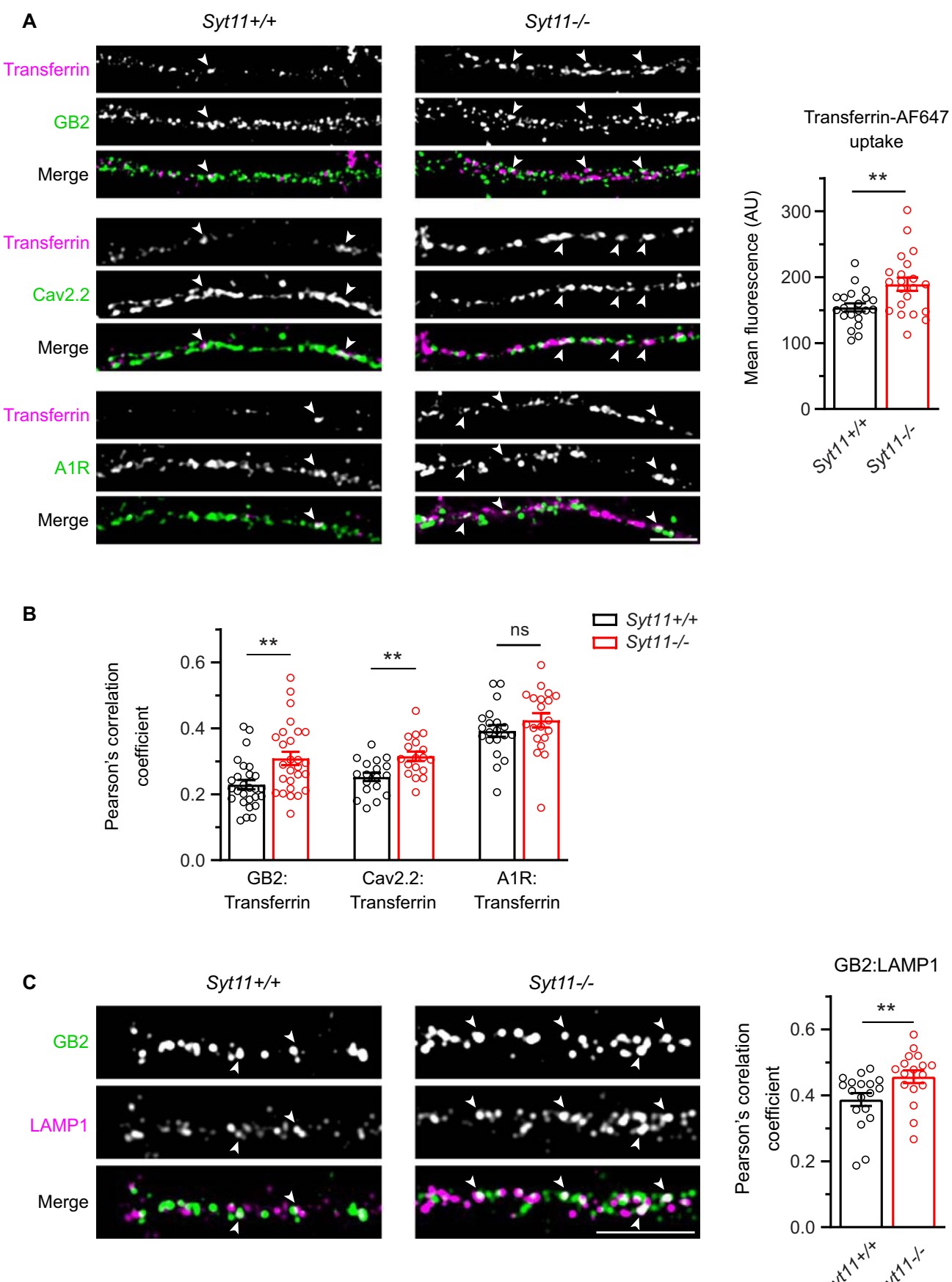

**Figure EV4. Syt11 stabilizes GBRs and Cav2.2 channels but not A1R at the cell surface of neurons.**

(A) Representative confocal images of dendrites of cultured $Syt11^{+/+}$ and $Syt11^{-/-}$ hippocampal neurons (DIV14). Neurons were incubated with Transferrin-AF647 (magenta) for 30 min to label early endosomes. Fixed and permeabilized neurons were then stained for endogenous GB2, Cav2.2, or A1R (all green). Arrowheads indicate examples of Transferrin-AF647+ vesicles carrying GB2, Cav2.2, or A1R. Scale bar, 5 μm. Increased Transferrin-AF647 uptake is observed in $Syt11^{-/-}$ compared to $Syt11^{+/+}$ neurons. $n = 21$ neurons for each genotype from 3 independent experiments. (B) Quantification of co-localization of GB2, Cav2.2, or A1R with Transferrin-AF647 in experiments described in (A). The Pearson's correlation coefficients indicate the degree of co-localization between Transferrin-AF647 and GB2, Cav2.2, or A1R in dendrites. In $Syt11^{-/-}$ neurons, co-localization with Transferrin-AF647 is increased for endogenous GB2 and Cav2.2, indicating increased internalization. GB2, $n = 27$ neurons; Cav2.2, $n = 18$ neurons; A1R, $n = 20$ neurons from 3 independent experiments. (C) Representative confocal images of dendrites of cultured $Syt11^{+/+}$ and $Syt11^{-/-}$ hippocampal neurons (DIV14) stained for endogenous GB2 (green) and the lysosome marker LAMP1 (magenta). Arrowheads indicate examples of co-localization of GB2 with LAMP1. Scale bar, 5 μm. Pearson's correlation coefficient indicates increased co-localization of GB2 with LAMP1 in dendrites of $Syt11^{-/-}$ neurons. $Syt11^{+/+}$, $n = 18$ neurons; $Syt11^{-/-}$, $n = 17$ neurons from 3 independent experiments. Data information: Data are presented as mean ± SEM. Statistical significance was determined by Welch's $t$-test (A), unpaired Student's $t$-test (B), or Mann–Whitney U test (C). ns not significant; $**p < 0.01$.

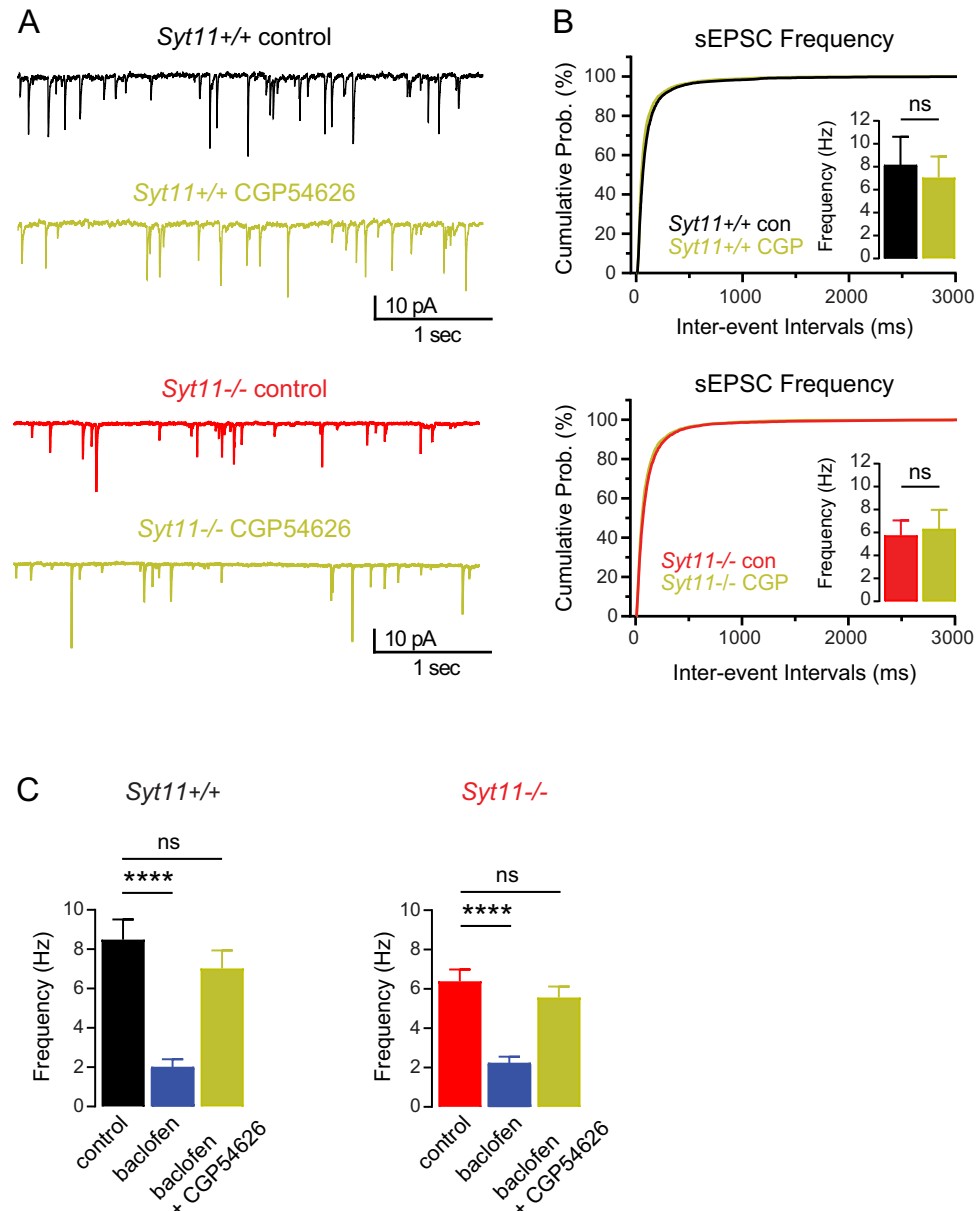

**Figure EV5. Lack of tonic or constitutive GBR activity in cultured *Syt11*[+/+] and *Syt11*[−/−] hippocampal neurons.**

(A) Representative traces of sEPSCs recorded from a *Syt11*[+/+] (top) and *Syt11*[−/−] (bottom) neuron in the presence of gabazine (10 μM) before (control, black/red) and after application of CGP54626 (4 μM, yellow). (B) Cumulative probability distributions of sEPSC inter-event intervals from *Syt11*[+/+] (top) and *Syt11*[−/−] (bottom) neurons recorded as in (A). In both genotypes, the sEPSC frequency (insets) was not significantly different in the presence of CGP54626 (CGP) compared to control (con). Upper inset: *Syt11*[+/+] neurons (con: 8.16 ± 2.46 Hz vs CGP: 7.07 ± 1.83 Hz). Lower inset: *Syt11*[−/−] neurons (con: 5.74 ± 1.32 Hz vs CGP: 6.31 ± 1.67 Hz). $n = 5$ neurons per genotype from 3 preparations. (C) Summary bar graph showing the sEPSC frequency of *Syt11*[+/+] (left) and *Syt11*[−/−] (right) neurons in the presence of gabazine (10 μM) before (control, black/red) and after application of baclofen (100 μM, blue) and baclofen + CGP54626 (4 μM, yellow). *Syt11*[+/+], $n = 11$ neurons; *Syt11*[−/−], $n = 16$ neurons from 6 preparations. Data information: Data are presented as mean ± SEM. Statistical significance was determined by Wilcoxon matched-pairs signed-rank test (B) or Friedman test and Dunn's multiple comparisons (C). ns not significant; ****$p < 0.0001$.

