## [Peer Review File · EMBO Reports]

Synaptotagmin-11 facilitates assembly of a presynaptic signaling complex in post-Golgi cargo vesicles

Luca Trovo, Stylianos Kouvaros, Jochen Schwenk, Diego Fernandez-Fernandez, Thorsten Fritzius, Pascal Rem, Simon Früh, Martin Gassmann, Bernd Fakler, Josef Bischofberger, and Bernhard Bettler

Corresponding author(s): Bernhard Bettler (bernhard.bettler@unibas.ch)

Review Timeline:

Submission Date:	18th Aug 23
Editorial Decision:	13th Sep 23
Revision Received:	22nd Mar 24
Editorial Decision:	9th Apr 24
Revision Received:	11th Apr 24
Accepted:	12th Apr 24

Transaction Report:

Dear Prof. Bettler,

Thank you for the submission of your research manuscript to our journal. We have now received the full set of referee reports that is copied below.

As you will see, the referees acknowledge that the findings are potentially interesting, but they also raise a number of concerns and have many suggestions on how to strengthen and verify your conclusions that need to be addressed.

Given these constructive comments, we would like to invite you to revise your manuscript with the understanding that the referee concerns (as detailed above and in their reports) must be fully addressed and their suggestions taken on board. Please address all referee concerns in a complete point-by-point response. Acceptance of the manuscript will depend on a positive outcome of a second round of review. It is EMBO Reports policy to allow a single round of revision only and acceptance or rejection of the manuscript will therefore depend on the completeness of your responses included in the next, final version of the manuscript.

We realize that it is difficult to revise to a specific deadline. In the interest of protecting the conceptual advance provided by the work, we recommend a revision within 3 months (December 13th). Please discuss the revision progress ahead of this time with the editor if you require more time to complete the revisions.

I am also happy to discuss the revision further via e-mail or a video call, if you wish.

*****IMPORTANT NOTE:

We perform an initial quality control of all revised manuscripts before re-review. Your manuscript will FAIL this control and the handling will be delayed IN CASE the following applies:

- 1) A data availability section providing access to data deposited in public databases is missing. If you have not deposited any data, please add a sentence to the data availability section that explains that.
- 2) Your manuscript contains statistics and error bars based on $n=2$. Please use scatter blots in these cases. No statistics should be calculated if $n=2$.

When submitting your revised manuscript, please carefully review the instructions that follow below. Failure to include requested items will delay the evaluation of your revision.*****

- 1) a .docx formatted version of the manuscript text (including legends for main figures, EV figures and tables). Please make sure that the changes are highlighted to be clearly visible.
- 2) individual production quality figure files as .eps, .tif, .jpg (one file per figure). Please download our Figure Preparation Guidelines (figure preparation pdf) from our Author Guidelines pages <https://www.embopress.org/page/journal/14693178/authorguide> for more info on how to prepare your figures.
- 3) a .docx formatted letter INCLUDING the reviewers' reports and your detailed point-by-point responses to their comments. As part of the EMBO Press transparent editorial process, the point-by-point response is part of the Review Process File (RPF), which will be published alongside your paper.
- 4) a complete author checklist, which you can download from our author guidelines (). Please insert information in the checklist that is also reflected in the manuscript. The completed author checklist will also be part of the RPF.
- 5) Please note that all corresponding authors are required to supply an ORCID ID for their name upon submission of a revised manuscript (). Please find instructions on how to link your ORCID ID to your account in our manuscript tracking system in our Author guidelines

()

6) We replaced Supplementary Information with Expanded View (EV) Figures and Tables that are collapsible/expandable online. A maximum of 5 EV Figures can be typeset. You have already implemented this formatting. But please note that the movies must be uploaded as a zipped file that contains the movie itself and the legend as README.txt file.

7) Please note that a Data Availability section at the end of Materials and Methods is now mandatory. In case you have no data that requires deposition in a public database, please state so. Please do not refer to "data is available upon request" in this section. See also .

Additional information on source data and instruction on how to label the files are available .

10) Figure legends and data quantification:

- the name of the statistical test used to generate error bars and P values,
- the number (n) of independent experiments (please specify technical or biological replicates) underlying each data point,
- the nature of the bars and error bars (s.d., s.e.m.)

- If the data are obtained from n {less than or equal to} 5, show the individual data points in addition to the SD or SEM.

- If the data are obtained from n {less than or equal to} 2, use scatter blots showing the individual data points.

11) Our journal encourages inclusion of *data citations in the reference list* to directly cite datasets that were re-used and obtained from public databases. Data citations in the article text are distinct from normal bibliographical citations and should directly link to the database records from which the data can be accessed. In the main text, data citations are formatted as follows: "Data ref: Smith et al, 2001" or "Data ref: NCBI Sequence Read Archive PRJNA342805, 2017". In the Reference list, data citations must be labeled with "[DATASET]". A data reference must provide the database name, accession number/identifiers and a resolvable link to the landing page from which the data can be accessed at the end of the reference. Further instructions are available at .

12) The "Declaration of Interests" should be renamed to our new 'Disclosure and competing interests statement'. For more information see

<https://www.embopress.org/page/journal/14693178/authorguide#conflictsofinterest>

13) All Materials and Methods need to be described in the main text. We would encourage you to use 'Structured Methods', our new Materials and Methods format. According to this format, the Materials and Methods section should include a Reagents and Tools Table (listing key reagents, experimental models, software and relevant equipment and including their sources and relevant identifiers) followed by a Methods and Protocols section in which we encourage the authors to describe their methods using a step-by-step protocol format with bullet points, to facilitate the adoption of the methodologies across labs.

More information on how to adhere to this format as well as downloadable templates (.doc or .xls) for the Reagents and Tools Table can be found in our author guidelines: <
<https://www.embopress.org/page/journal/14693178/authorguide#manuscriptpreparation>>. An example of a Method paper with Structured Methods can be found here: .

14) As part of the EMBO publication's Transparent Editorial Process, EMBO Reports publishes online a Review Process File to accompany accepted manuscripts. This File will be published in conjunction with your paper and will include the referee reports, your point-by-point response and all pertinent correspondence relating to the manuscript.

Yours sincerely,

Referee #1:

This is a very nice study, from an excellent group, linking CaV2.2, GABA-B receptors and Synaptotagmin11 in post Golgi trafficking complexes involved particularly in axonal trafficking. It is, for the most part, very well controlled using knockout tissue where possible.

I have a few comments for improvements.

- 1) P4, line 1 should read CACNA1B not CAC1B
- 2) P4 lines 14-19. The authors have shown that G α signaling is not affected by binding to Syt11, but what about G $\beta\gamma$ signaling, which is more relevant to CaV2.2?
- 3) Does the interaction between the N- and C-terminal fragments of fluorescent Venus protein in any way promote an interaction between the proteins to which they are tagged, for example by stabilizing the complex with KCTD16?
- 4) Has the rabbit anti-Cav2.2 (Cat# AB5154, Millipore, 1:1000) been verified against CaV2.2 knockout tissue? If not, there may be some major questions that need to be answered about the specificity of this antibody.
- 5) Page 6 line 26 onwards. I note the authors carefully mention CaV channels in this physiology section (line 37). Although CaV2.2 is more inhibited by G $\beta\gamma$ -mediated signaling, inhibition of CaV2.1 also occurs, and this channel is strongly expressed presynaptically, and also mediates release. Experiments should be included in the physiology section to determine what proportion of the signal is due to CaV2.2, by using ω -conotoxin GVIA.
- 6) Is there any evidence that the same trafficking processes described in this study are also occurring for CaV2.1? Comment should be made on this.
- 7) Page 7, line 19. Please quantify the statement "slightly more frequent".
- 8) Page 8, line 2 typo should be concentration.
- 9) Page 8, line 21. the calcium channel pore-forming subunit is $\alpha 1$ not α . Please also mention here the importance of the $\alpha 2\delta$ as well as β in the complex, for trafficking these channels.

Referee #3:

The study by Trovo et al. follows the identification of GABABR interacting molecules, which has been in focus of the participating labs since almost two decades. The concepts that G-protein coupled receptors should be tightly linked to their effectors to have the precision in signalling as we know from physiological readouts is one of the very strong contributions. The employed

proteomic approaches did resolve key molecular components that are responsible for the spatiotemporal confinement of the Cav/GABAB signalling complexes.

In line with this general concept the major message of the paper by Troco et al. is that the calcium channel modulation by GABAB receptors is based on the preassembly of signalling complex after the production of the different components in the endoplasmic reticulum (ER). The where and when the core complex of heteromeric GABAB receptors, the auxiliary KCTD16 protein and the Cav2.2 channels are assembled is still a question. The meeting point could be in the synapse but could also well be on the way to the synapse by meeting and connecting these proteins within a specific vesicular compartment. Proteomic studies by Müller et al. 2010 and Schwenk et al. 2016 have suggested Synaptotagmin 11 (Syt11) as a significant binding partner for Cav2.2 channels and GABAB receptors. In the current study this data from proteomic studies have been re-evaluated and confirmed (Fig.1) and further explored by the identification where and when Synaptotagmin11 binds to both KCTD16 and Cav2.2. Labelling of postsynaptic scaffold protein PSD95 and labelling of GB2 and Syt11 with one half of split-GFP was used to identify whether the association between GABAB receptor and Syt11 is present in the synapse or close by. Here a low number of spines along the dendrite does indeed suggest the tight association between both partners. These data are based on the analysis of 22 spines out of three experiments (Fig.2 D). I wonder how robust this essay will be given the fact that even in culture neurons (14-15 DIV) receive thousands of synaptic inputs. An increase in the number of analysed spines would really strengthen this point, otherwise it smells that these data are very subjective to assign the Syt11-GABAB receptor complex to the spine neck.

Live imaging and use of markers for endosomes and dense core vesicles (DCV) in combination with SIM microscopy of fixed samples allowed to identify that indeed a fraction of DCV seem to contain GABAB receptors and Cav2.2 channels, which is dependent on the expression of Syt11. The fast dynamic of the vesicles suggests active trafficking of the preformed complex of Cav2.2 channels and GABABR inside the cellular compartments. These data support the idea that GABABR and Cav2.2 channels find each other along the trafficking pathway and are packed in specific vesicles to reach dendrites, axons and potentially the presynaptic membrane. In a next set of experiments the authors started to test the functional impact of the preformed complex between GABAB receptor, KCTD16, Cav2.2 and Syt11.

How the disturbance of the identified trafficking complex composed of Syt11, GABABR and Cav2.2 channels interfere with synaptic transmission and network activity is not clear and lacks many controls. The first data in Fig.4 A-C demonstrate the impact of Syt11 on the frequency of spontaneous transmitter release and allow to assign the major functional impact of Syt11 to the presynaptic release probability. Whether this modulation has direct impact on the bursting activity in neuronal networks is not clear. This is probably based on a misinterpretation of bursting activity in primary cultures, which is a phenomenon that is heavily investigated in cultures but has no direct link to the here proposed early network oscillations (ENOs) seen in *in vivo* data (see Corner 2008). To compare the bursting activity with ENOs, as reported from experiments in acute slices from young rats (Garaschuk et al. 1998) is completely misleading. These events are based on the excitatory function of GABAA receptors in young neuronal networks from rats at the postnatal days P1-P5/6. Similar calcium bursts can be seen in young cultures in the first week *in vitro* but disappear after the so-called GABA switch, when GABAA receptors action is inhibitory (Ganguly et al., 2001), similar to the development *in vivo*. The events described in the study by Trovo et al. are recorded from neurons that have been cultured 14-15 DIV and do represent sodium channel activity triggered bursts of action potential rather than calcium bursts reported by Garaschuk et al. 1998. The time window to observe ENOs in slices or in primary cultures is within the first postnatal week or until 10-11 DIV (Garaschuk et al. 1998, Ganguly et al. 2001). I would argue that the here seen bursts represent rather the stereotypic activity of neuronal networks *in vitro* that lack sensory input and can arrest in such bursting behaviour for very long time after the initial synaptogenesis (DIV 10-14) *in vitro* (Corner, 2008). In addition to this wrong interpretation, I wonder how robust the change in frequency of such bursting activity depends on the absence or presence of Syt11? It could well be just a variability in the bursting activity of different neuronal cultures. To rule this out I would control whether at different timepoints of network development or in sister cultures from the same preparation the slight difference in the burst frequency resist. Based on the fact that network bursting is linked to synaptic activity I wonder why the investigators did not probe their proposed decrease of Cav2.2 and GABAB receptors surface expression by the use of specific pharmacological tools?

Using baclofen to activate the surface population of GABAB receptors does indeed indicate that sEPSC frequency modulation depend on the expression of Syt11 and impact on the expression of GABAB receptors (Fig.4 G-I). But how this aspect is related to the expression of Cav2.2 channels in the synapse is not addressed at all. Here I would like to make a suggestion why this control could matter. As pointed out in the discussion, the authors are aware that the diversity of Cav2 channel isoforms is a well-known variable to diversify the release probability of synapses. Hence to probe whether the mix of Cav2 isoforms inside synapses differ between Syt11 *+/+* and Syt11 *-/-* synapses should be tested before arguing that you indeed see a downregulation of Cav2.2 channels. There are excellent tools to do this by employing presynaptic targeted and genetic encoded calcium sensors in combination with highly specific blockers of Cav2 isoforms, see for example (Brockhaus et al., 2019). Another aspect related to the firm connection between Cav2.2 channels and GABAB receptors is the abundance of Cav2 isoforms in the presynaptic terminals. Previous studies have shown that overexpression of Cav2.2 channels can wipe out the population of Cav2.1 channels, whereas overexpression of Cav2.1 channels cannot (Cao and Tsien, 2010). If indeed Syt11 has an impact on the traffic and GABABR stabilize Cav2.2 channels much stronger in the synapse than Cav2.1 channels one would expect clear differences in the accumulation of Cav2 channel isoforms inside the synapse when indeed Syt11 is a specific trafficking partner of Cav2.2 channels.

The experiments in Fig.4 J-L seem to suggest such a situation but also here I would recommend more controls. For example, the tonic activity of GABABR may is affected dependent on the composition of the signalling complex. May you can use specific antagonists for GABABR or GABA transporters as employed in previous studies (Laviv et al., 2010)?

Based on such experiments you may can come up with a more direct interpretation of the bursting activity. As we know from several identified synapses there is not always a direct correlation between just the number of calcium channels and their impact

on synaptic release probability see (Aldahabi et al., 2022; Eltes et al., 2017; Rebola et al., 2019).

In summary, in my opinion the paper needs some further characterisation of the proposed functional consequences in addition to the subcellular identification of the complexes as mentioned above. The molecular interactions are confirmed several times and it would be interesting whether indeed GABABR together with Syt11 determine the population size and stability of Cav2.2 channels in individual synapses.

References:

- Aldahabi, M., Balint, F., Holderith, N., Lorincz, A., Reva, M., and Nusser, Z. (2022). Different priming states of synaptic vesicles underlie distinct release probabilities at hippocampal excitatory synapses. *Neuron* 110, 4144-4161.e4147.
- Brockhaus, J., Brügger, B., and Missler, M. (2019). Imaging and Analysis of Presynaptic Calcium Influx in Cultured Neurons Using synGCaMP6f. *Frontiers in Synaptic Neuroscience* 11.
- Cao, Y.Q., and Tsien, R.W. (2010). Different relationship of N- and P/Q-type Ca²⁺ channels to channel-interacting slots in controlling neurotransmission at cultured hippocampal synapses. *J Neurosci* 30, 4536-4546.
- Corner, M.A. (2008). Spontaneous neuronal burst discharges as dependent and independent variables in the maturation of cerebral cortex tissue cultured in vitro: a review of activity-dependent studies in live 'model' systems for the development of intrinsically generated bioelectric slow-wave sleep patterns. *Brain Res Rev* 59, 221-244.
- Eltes, T., Kirizs, T., Nusser, Z., and Holderith, N. (2017). Target Cell Type-Dependent Differences in Ca²⁺ Channel Function Underlie Distinct Release Probabilities at Hippocampal Glutamatergic Terminals. *J Neurosci* 37, 1910-1924.
- Ganguly, K., Schinder, A.F., Wong, S.T., and Poo, M. (2001). GABA itself promotes the developmental switch of neuronal GABAergic responses from excitation to inhibition. *Cell* 105, 521-532.
- Laviv, T., Riven, I., Dolev, I., Vertkin, I., Balana, B., Slesinger, P.A., and Slutsky, I. (2010). Basal GABA regulates GABA(B)R conformation and release probability at single hippocampal synapses. *Neuron* 67, 253-267.
- Rebola, N., Reva, M., Kirizs, T., Szoboszlay, M., Lorincz, A., Moneron, G., Nusser, Z., and DiGregorio, D.A. (2019). Distinct Nanoscale Calcium Channel and Synaptic Vesicle Topographies Contribute to the Diversity of Synaptic Function. *Neuron*.

Please find below our point-by-point response to the reviewers' comments in blue. The changes we made to the manuscript are indicated in red.

Referee #1:

Query 1. P4, line 1 should read CACNA1B not CAC1B

The gene symbol CACNA1B corresponds to the UniProt entry name CAC1B. For clarity, we have replaced CAC1B with the subunit name $\alpha 1B$. This modification has been implemented on page 3 and in the table in Fig 1 containing the proteomic data.

P3, I34.

Ten of the Syt11-interacting proteins were previously identified in anti-GBR APs [9, 10, 16], including GB1, GB2, KCTD16 and the Cav2.2 (N-type voltage-gated Ca^{2+} channel) subunit $\alpha 1B$.

Query 2. P4 lines 14-19. The authors have shown that $G\alpha$ signaling is not affected by binding to Syt11, but what about $G\beta\gamma$ signaling, which is more relevant to CaV2.2?

To examine the potential modulation of $G\beta\gamma$ signaling of GBRs by Syt11, we used a BRET assay reporting $G\beta\gamma$ release from the activated G protein. The BRET changes induced by application of GABA to HEK293T cells co-expressing GBRs, KCTD16 and the BRET sensors were not significantly different in the presence or absence of Syt11. Furthermore, the BRET assay indicated that Syt11 does not induce constitutive receptor activity, as also observed with $G\alpha$ signaling, addressing concerns raised in Query 5 by Reviewer 3. The results of these BRET experiments are presented in panel B of the new Figure 2. We additionally revised the Methods section to provide details about the BRET assay.

P4, I19.

Luciferase activity at baseline was not significantly different in the presence or absence of Syt11, supporting that Syt11 does not influence constitutive $G\alpha$ signaling of GBRs (Fig 2A). We also addressed whether Syt11 modulates $G\beta\gamma$ signaling of GBRs using a BRET assay based on the binding of Venus-tagged $G\beta\gamma$ to a membrane-associated GRK3ct-luciferase (masGRK3ct-NanoLuc) [32, 33]. The BRET change induced by the application of 100 μM GABA to cells co-expressing GBRs, KCTD16, and the BRET sensors was not significantly different in the presence or absence of Syt11 (Fig 2B). After blocking GBRs with 4 μM CGP54626, an inverse agonist inhibiting constitutive GBR activity [34], the BRET signal returned to baseline without undershooting, irrespective of the presence of Syt11 (Fig 2B). The data from CGP54626 experiments corroborate that Syt11 does not induce constitutive receptor activity. Collectively, these biochemical experiments support the conclusion that Syt11 does not influence $G\alpha$ or $G\beta\gamma$ signaling of GBRs.

P11, I31.

Expression of Syt11 does not significantly change basal and GABA-induced luciferase activity. Non-linear regression curve fits of $n = 8$ independent experiments per condition. Mean \pm SEM, $p = 0.5929$, extra sum-of-squares F-test (middle). Baseline activity (BL), $p = 0.5882$, unpaired t-test (bottom).

(B) $G\beta\gamma$ released upon GBR activation in HEK293T cells was monitored using a BRET assay reporting the binding of Venus-tagged $G\beta\gamma$ to a membrane-associated GRK3ct-luciferase (top). Representative experiments (middle) and quantification of BRET changes (bottom) induced by the application of GABA and the inverse agonist CGP54626. Co-expression of Syt11 does not significantly alter GABA ($p = 0.2068$) and CGP54626 ($p = 0.9777$) induced BRET changes compared to control (unpaired Student's t test). Mean \pm SEM from $n = 8$ independent experiments recorded in triplicates, ns = not significant.

P22, I19.

To monitor G $\beta\gamma$ signaling by GBRs, HEK293T cells were transiently transfected with Flag-GB1b, Flag-GB2, KCTD16, GRK3ct-RLuc [32, 33] G α , G β 2, and Venus-G γ ₂ plasmids, with or without Syt11. Transfected cells were seeded into 96-well microplates (Greiner Bio-One) at a density of 100,000 cells/well. After 18 hrs, cells were washed, and coelenterazine h (5 μ M, NanoLight Technologies, Prolume Ltd., Pinetop-Lakeside, United States of America) added for 5 min. Luminescence and fluorescence signals were alternately recorded for a total duration of 640 sec using a Spark[®] microplate reader. GABA and CGP54626 were injected at 127 and 384 seconds, respectively, using the Spark[®] microplate reader injection system. The BRET ratio was calculated as the ratio of the light emitted by Venus-G γ ₂ (530 – 570 nm) over the light emitted by GRK3ct-RLuc (370 – 470 nm). BRET ratios were adjusted by subtracting the ratios obtained when RLuc fusion proteins were expressed alone. Each data point represents a technical triplicate.

Query 3. Does the interaction between the N- and C-terminal fragments of fluorescent Venus protein in any way promote an interaction between the proteins to which they are tagged, for example by stabilizing the complex with KCTD16?

We found that comparable amounts of Syt11 and KCTD16 proteins co-immunoprecipitated with GB2 from membranes of transfected HEK293T cells, irrespective of whether proteins tagged with Venus fragments (HA-GB2-VC, Syt11-VN) or proteins without Venus tags (HA-GB2, Syt11) were co-expressed with KCTD16. The data support that the VN- and VC-tags on Syt11 and GB2 neither promote an interaction between the two proteins nor stabilize the complex with KCTD16. The results from a representative experiment, along with quantification from five independent experiments, are presented in the new Fig EV2B.

P5, I5.

The Venus fragment tags on GB2 and Syt11 did not impede nor enhance KCTD16-mediated complex formation, as demonstrated by co-AP of comparable amounts of Syt11 and KCTD16 with GB2 from membranes of transfected HEK293 cells, both in the absence and presence of the Venus fragment tags (Fig EV2B).

P16, I1.

(B) Representative Western blots (left) and corresponding quantifications (right) of APs with anti-HA antibodies from cell lysates of transfected HEK293T cells expressing the indicated constructs. AP and input lanes were probed with anti-Syt11 (top), anti-KCTD16 (middle), and anti-HA (bottom) antibodies. The presence of VN- or VC-tags on Syt11 and GB2, respectively, does not significantly alter the amounts of KCTD16 ($p = 0.436$) and Syt11 ($p = 0.858$) co-purified with GB2. Values are presented as mean \pm SEM, ns = not significant, unpaired t-test.

Query 4. Has the rabbit anti-Cav2.2 (Cat# AB5154, Millipore, 1:1000) been verified against CaV2.2 knockout tissue? If not, there may be some major questions that need to be answered about the specificity of this antibody.

We exclusively used the rabbit anti-Cav2.2 antibody from Millipore (Cat# AB5154) to detect the heterologously expressed α 1B subunit in HEK293T cells (Fig 1B and C). In these experiments, the antibody produced only a very weak non-specific background signal in non-transfected cells, validating its suitability for this application. For imaging endogenous α 1B protein in neurons, we always used the rabbit anti-Cav2.2 antibody from Alomone Labs (Cat# ACC-002, RRID:AB_2039766). This antibody has been verified against Cav2.2 knockout tissue.

P11, I18.

The $\alpha 1B$ subunit of Cav2.2 channels, co-expressed with auxiliary β and $\alpha 2\delta$ subunits, was identified on Western blots using the anti-Cav2.2 antibody from Millipore (# AB5154).

P13, I21.

For immunochemistry, the endogenous $\alpha 1B$ subunit of Cav2.2 channels was detected using the anti-Cav2.2 antibody from Alamone Labs (#ACC-002, RRID:AB2039766), which was validated with *CNCNA1b*^{-/-} mouse tissue [85].

Query 5. Page 6 line 26 onwards. I note the authors carefully mention CaV channels in this physiology section (line 37). Although CaV2.2 is more inhibited by G $\beta\gamma$ -mediated signaling, inhibition of CaV2.1 also occurs, and this channel is strongly expressed presynaptically, and also mediates release. Experiments should be included in the physiology section to determine what proportion of the signal is due to CaV2.2, by using ω -conotoxin GVIA.

As recommended by the referee, we assessed the relative contribution of Cav2.2 and Cav2.1 channels to release in cultured hippocampal neurons of *Syt11*^{+/+} and *Syt11*^{-/-} mice by recording the sEPSC frequencies in the presence of ω -conotoxin and ω -agatoxin. The corresponding data have been incorporated into the Results section and are presented in a new Figure 6.

P8, I1.

Hippocampal *Syt11*^{-/-} neurons exhibit a significant deficit in presynaptic Cav2.2 channels

Various types of Cav channels contribute to activity-dependent neurotransmitter release at brain synapses, with Cav2.1 and Cav2.2 channels being most prominent [55-58]. To determine the contribution of Cav2.2 channels to synaptic transmitter release in *Syt11*^{-/-} neurons and explore potential compensatory changes in Cav2.1 channels, we measured sEPSC frequencies in the consecutive presence of ω -conotoxin (blocking Cav2.2 channels), ω -conotoxin + ω -agatoxin (blocking Cav2.1 channels), and ω -conotoxin + ω -agatoxin + TTX (preventing action potential-dependent release) (Fig 6A). Blocking Cav2.2 channels by ω -conotoxin significantly reduced the frequency of sEPSCs and shifted the distribution of inter-event intervals toward larger values in both *Syt11*^{-/-} and *Syt11*^{+/+} neuronal cultures (Fig 6B and C). However, ω -conotoxin was significantly less efficient in inhibiting the sEPSC frequency in *Syt11*^{-/-} neurons compared to *Syt11*^{+/+} neurons (Fig 6D), consistent with a reduction in presynaptic Cav2.2 channels. Notably, the combined inhibitory effect of ω -agatoxin and TTX, but not of ω -agatoxin alone, was significantly larger in *Syt11*^{-/-} neurons when applied after ω -conotoxin (Fig 6D and E). Altogether, this indicates a reduction in the relative contribution of presynaptic Cav2.2 channels to synaptic release in *Syt11*^{-/-} neurons, partially offset by an upregulation of other Cav channels. Additionally, the lack of significant differences in TTX-insensitive mEPSCs between genotypes (Fig 6D) indicates comparable synapse density.

P9, I5.

However, subtype-selective Cav channel blockers revealed a significant reduction in presynaptic Cav2.2 channels in *Syt11*^{-/-} neurons, which are not fully compensated for by other Cav channels. This is consistent with the finding that Syt11 exhibits a preference for binding to Cav2.2 channels over Cav2.1 channels (this study), and that presynaptic slots accepting Cav2.2 channels reject Cav2.1 channels [57].

P14. I25.

Figure 6 - Cultured *Syt11*^{-/-} hippocampal neurons exhibit a deficit in presynaptic Cav2.2 channels

(A) Representative traces of sEPSCs from *Syt11*^{+/+} and *Syt11*^{-/-} hippocampal neurons in culture recorded at DIV15-19 in the presence of gabazine (10 μ M) before (control, black/red) and after application of ω -conotoxin (1 μ M, blue), ω -conotoxin + ω -agatoxin (500 nM, yellow) and ω -conotoxin + ω -agatoxin + TTX (1 μ M, magenta).

(B) Cumulative probability distributions of sEPSC inter-event intervals of *Syt11^{+/+}* and *Syt11^{-/-}* neurons recorded as in (A).

(C) Summary bar graph depicting the sEPSC frequency of *Syt11^{+/+}* and *Syt11^{-/-}* neurons recorded as in (A). In both genotypes, the frequency of sEPSCs was significantly reduced by the application of ω -conotoxin, ω -conotoxin + ω -agatoxin, and ω -conotoxin + ω -agatoxin + TTX. Wilcoxon test; α -values were adjusted by the Bonferroni correction for multiple comparisons. n = 9-11 neurons per genotype from 4 preparations.

(D) Summary bar graph depicting the percentage inhibition of sEPSC frequency by ω -conotoxin, ω -agatoxin, and TTX in cultured hippocampal neurons of *Syt11^{+/+}* (black) and *Syt11^{-/-}* mice (red). Inhibition by ω -conotoxin (blocking Cav2.2 channels) is significantly reduced in *Syt11^{-/-}* compared to *Syt11^{+/+}* neurons (*Syt11^{+/+}*: 60.55 ± 4.42 % vs *Syt11^{-/-}*: 31.90 ± 7.80 %, p = 0.005, Mann-Whitney U test). The ω -agatoxin-sensitive (blocking Cav2.1 channels) and TTX-sensitive components of inhibition, as well as the TTX-insensitive component (mEPSCs) show no significant difference between genotypes.

(E) The combined ω -agatoxin- and TTX-sensitive component of inhibition is significantly increased in *Syt11^{-/-}* compared to *Syt11^{+/+}* neurons (*Syt11^{+/+}*: 29.33 ± 4.64 % vs *Syt11^{-/-}*: 47.78 ± 7.37 %, p = 0.038, Mann-Whitney U test).

Values in bar graphs are presented as mean \pm SEM, n = 9-11 neurons per genotype from 4 preparations. Statistical significance is indicated as *p < 0.05, **p < 0.01, ns = not significant. Source data are available online for this figure.

Query 6. Is there any evidence that the same trafficking processes described in this study are also occurring for Cav2.1? Comment should be made on this.

Mass spectrometry-based proteomic data showed that the Cav2.1 α 1A subunit fails to co-immunoprecipitate with Syt11 from mouse brain membrane extracts (Figure 1A), supporting that Cav2.1 channels are not constituents of GBR/Syt11 complexes. Additionally, we now conducted co-immunoprecipitation experiments in transfected HEK293T cells, revealing that significantly more Syt11 co-purifies with Cav2.2 α 1B protein than Cav2.1 α 1A protein (Fig EV1C). Biochemical data, therefore, support that mainly Cav2.2 channels are constituents of GBR/Syt11 complexes. Similarly, the pharmacological experiments suggested by the referees now demonstrate that *Syt11^{-/-}* neurons primarily lack presynaptic Cav2.2 channels (Fig 6). We discuss these findings in the revised version of the manuscript.

P4, I8.

Notably, the α 1A subunit of Cav2.1 channels (P/Q-type voltage-gated Ca²⁺ channels) failed to co-purify with Syt11 from mouse brains (Fig 1A). Additionally, the α 1A subunit exhibited a significantly decreased propensity to co-purify Syt11 in APs from transfected HEK293T cells compared to the α 1B subunit of Cav2.2 channels (Fig EV1C).

P9, I5.

However, subtype-selective Cav channel blockers revealed a significant reduction in presynaptic Cav2.2 channels in *Syt11^{-/-}* neurons, which are not fully compensated for by other Cav channels. This is consistent with the finding that Syt11 exhibits a preference for binding to Cav2.2 channels over Cav2.1 channels (this study), and that presynaptic slots accepting Cav2.2 channels reject Cav2.1 channels [57].

P15, I25.

(C) Significantly increased co-purification of Myc-tagged Syt11 with the α 1B subunit of Cav2.2 channels compared to α 1A subunit of Cav2.1 channels from total cell lysates of transfected HEK293T cells. Auxiliary β and α 2 δ subunits were co-expressed with the α 1A and α 1B subunits. Representative Western blots (left) and quantification from n = 4 independent experiments (right). Values are presented as mean \pm SEM, *p = 0.028, Mann-Whitney U test.

Query 7. Page 7, line 19. Please quantify the statement "slightly more frequent".

We have now quantified the frequency of small amplitude events in *Syt11*^{-/-} neuronal cultures in the presence and absence of baclofen and found no statistically significant difference (Fig 5G and H). We therefore have removed the statement “indicating that baclofen converts large amplitude events to small amplitude events”.

P7, I23.

Plotting the number of sEPSC against EPSC amplitudes revealed that in *Syt11*^{+/+} neuronal cultures, baclofen significantly reduced both large and small amplitude events (Fig 5G). In contrast, in *Syt11*^{-/-} cultures, baclofen failed to significantly reduce small amplitude events (Fig 5H). This indicates a reduction in the number of synapses exhibiting GBR-mediated inhibition of glutamate release in the absence of Syt11.

P14. I15.

(G) Number of sEPSCs plotted against sEPSC amplitudes in *Syt11*^{+/+} neurons in the presence and absence (con) of baclofen (left). Activation of GBRs significantly inhibits small amplitude (<10 pA) and large amplitude (>10 pA) events (right).

(H) Number of sEPSCs plotted against sEPSC amplitudes in *Syt11*^{-/-} neurons in the presence and absence (con) of baclofen (left). Activation of GBRs significantly inhibits large amplitude (>10 pA) but not small amplitude (<10 pA) events.

Query 8. Page 8, line 2 typo should be concentration.

The typo has been corrected.

Query 9. Page 8, line 21. the calcium channel pore-forming subunit is $\alpha 1$ not α . Please also mention here the importance of the $\alpha 2\delta$ as well as β in the complex, for trafficking these channels.

We have corrected the nomenclature and now emphasize the role of the β and $\alpha 2\delta$ subunits in channel trafficking.

P9, I20.

KCTD proteins associate with GBRs at the ER membrane [14], where the intracellular β subunit of the Cav channel assembles with the pore-forming $\alpha 1$ subunit [68]. Assembly with the β subunit protects the $\alpha 1$ subunit from ER-associated degradation, facilitating the forward trafficking of the channel complex [68, 69]. Since we observe a reduced co-localization of GBRs and Cav2.2 in post-Golgi vesicles of *Syt11*^{-/-} neurons, we propose that Syt11 recruits Cav2.2 channel subunits together with KCTD16-bound GBRs from the trans-Golgi network into vesicles. Efficient trafficking of Cav channels to release sites requires proteolytic processing of a single precursor protein into a disulfide-bonded $\alpha 2\delta$ subunit along the biosynthetic pathway [70-72]. Notably, $\alpha 2\delta$ co-purified with the $\alpha 1B$ subunit in both anti-GBR and anti-KCTD16 APs [10]. This suggests that all components of GBR/Cav2.2 signaling complexes assemble in Syt11 vesicles. Subsequently, these vesicles process and traffic the signaling complex to presynaptic sites.

Referee #3:

Query 1. The where and when the core complex of heteromeric GABAB receptors, the auxiliary KCTD16 protein and the Cav2.2 channels are assembled is still a question. The meeting point could be in the synapse but could also well be on the way to the synapse by meeting and connecting these proteins within a specific vesicular compartment.

Our prior research indicates that KCTD16 assembles with the GBR in the ER - where the $\alpha 1$ and β subunits of Cav channels also come together. The assembly site for $\alpha 2\delta$ with Cav channels remains unknown, however, efficient trafficking of Cav channels to release sites

requires the proteolytic processing of the $\alpha 2\delta$ precursor along the biosynthetic pathway. Notably, $\alpha 2\delta$ co-purified with the $\alpha 1B$ subunit in both anti-GBR and anti-KCTD16 APs. This suggests that all components of GBR/Cav2.2 signaling complexes assemble in Syt11 vesicles. This is consistent with our observation that a reduced co-localization of GBRs and Cav2.2 in post-Golgi vesicles of *Syt11*^{-/-} neurons results in impaired GBR-mediated inhibition of spontaneous neurotransmitter release. In the revised manuscript this is now discussed in more detail.

P9, I20.

KCTD proteins associate with GBRs at the ER membrane [14], where the intracellular β subunit of the Cav channel assembles with the pore-forming $\alpha 1$ subunit [68]. Assembly with the β subunit protects the $\alpha 1$ subunit from ER-associated degradation, facilitating the forward trafficking of the channel complex [68, 69]. Since we observe a reduced co-localization of GBRs and Cav2.2 in post-Golgi vesicles of *Syt11*^{-/-} neurons, we propose that Syt11 recruits Cav2.2 channel subunits together with KCTD16-bound GBRs from the trans-Golgi network into vesicles. Efficient trafficking of Cav channels to release sites requires proteolytic processing of a single precursor protein into a disulfide-bonded $\alpha 2\delta$ subunit along the biosynthetic pathway [70-72]. Notably, $\alpha 2\delta$ co-purified with the $\alpha 1B$ subunit in both anti-GBR and anti-KCTD16 APs [10]. This suggests that all components of GBR/Cav2.2 signaling complexes assemble in Syt11 vesicles. Subsequently, these vesicles process and traffic the signaling complex to presynaptic sites.

Query 2. Labelling of postsynaptic scaffold protein PSD95 and labelling of GB2 and Syt11 with one half of split-GFP was used to identify whether the association between GABAB receptor and Syt11 is present in the synapse or close by. Here a low number of spines along the dendrite does indeed suggest the tight association between both partners. These data are based on the analysis of 22 spines out of three experiments (Fig.2 D). I wonder how robust this essay will be given the fact that even in culture neurons (14-15 DIV) receive thousands of synaptic inputs. An increase in the number of analysed spines would really strengthen this point, otherwise it smells that these data are very subjective to assign the Syt11-GABAB receptor complex to the spine neck.

We have increased the number of spines included in this analysis, with data from the examination of 83 spines now presented in Figure 3D. The additional results confirm that the BiFC complex is mostly absent from the spine head. However, they also reveal a broader distribution at the base of the spine neck compared to the previous dataset. It is important to note that our analysis was restricted to mushroom-shaped and stubby spines testing positive for both the Syt11-VN/GB2-VC BiFC and PSD95.

P12, I24.

A total of n = 83 spines were analyzed (4 independent preparations).

Query 3. How the disturbance of the identified trafficking complex composed of Syt11, GABABR and Cav2.2 channels interfere with synaptic transmission and network activity is not clear and lacks many controls. The first data in Fig.4 A-C demonstrate the impact of Syt11 on the frequency of spontaneous transmitter release and allow to assign the major functional impact of Syt11 to the presynaptic release probability. Whether this modulation has direct impact on the bursting activity in neuronal networks is not clear. This is probably based on a misinterpretation of bursting activity in primary cultures, which is a phenomenon that is heavily investigated in cultures but has no direct link to the here proposed early network oscillations (ENOs) seen in in vivo data (see Corner 2008). To compare the bursting activity with ENOs, as reported from experiments in acute slices from young rats (Garaschuk et al. 1998) is completely misleading. These events are based on the excitatory function of GABAA receptors in young neuronal networks from rats at the postnatal days P1-P5/6. Similar calcium bursts can be seen in young cultures in the first week in vitro but disappear after the so-called GABA switch, when GABAA receptors action is inhibitory (Ganguly et al.,

2001), similar to the development *in vivo*. The events described in the study by Trovo et al. are recorded from neurons that have been cultured 14-15 DIV and do represent sodium channel activity triggered bursts of action potential rather than calcium bursts reported by Garaschuk et al. 1998. The time window to observe ENOs in slices or in primary cultures is within the first postnatal week or until 10-11 DIV (Garaschuk et al. 1998, Ganguly et al. 2001). I would argue that the here seen bursts represent rather the stereotypic activity of neuronal networks *in vitro* that lack sensory input and can arrest in such bursting behaviour for very long time after the initial synaptogenesis (DIV 10-14) *in vitro* (Corner, 2008). In addition to this wrong interpretation, I wonder how robust the change in frequency of such bursting activity depends on the absence or presence of Syt11? It could well be just a variability in the bursting activity of different neuronal cultures. To rule this out I would control whether at different timepoints of network development or in sister cultures from the same preparation the slight difference in the burst frequency resist.

We agree with the referee that a link between the observed bursting activity in neuronal cultures and *in vivo* ENOs is speculative. Given the unclear physiological significance of the observed bursting activity in neuronal cultures and the fact that this aspect is not a primary focus of our work, we have decided to remove the associated data (previously presented in Fig 4, panels D-F) from our manuscript.

Query 4. Based on the fact that network bursting is linked to synaptic activity I wonder why the investigators did not probe their proposed decrease of Cav2.2 and GABAB receptors surface expression by the use of specific pharmacological tools? Using baclofen to activate the surface population of GABAB receptors does indeed indicate that sEPSC frequency modulation depend on the expression of Syt11 and impact on the expression of GABAB receptors (Fig.4 G-I). But how this aspect is related to the expression of Cav2.2 channels in the synapse is not addressed at all. Here I would like to make a suggestion why this control could matter. As pointed out in the discussion, the authors are aware that the diversity of Cav2 channel isoforms is a well-known variable to diversify the release probability of synapses. Hence to probe whether the mix of Cav2 isoforms inside synapses differ between Syt11 *+/+* and Syt11 *-/-* synapses should be tested before arguing that you indeed see a downregulation of Cav2.2 channels. There are excellent tools to do this by employing presynaptic targeted and genetic encoded calcium sensors in combination with highly specific blockers of Cav2 isoforms, see for example (Brockhaus et al., 2019). Another aspect related to the firm connection between Cav2.2 channels and GABAB receptors is the abundance of Cav2 isoforms in the presynaptic terminals. Previous studies have shown that overexpression of Cav2.2 channels can wipe out the population of Cav2.1 channels, whereas overexpression of Cav2.1 channels cannot (Cao and Tsien, 2010). If indeed Syt11 has an impact on the traffic and GABABR stabilize Cav2.2 channels much stronger in the synapse than Cav2.1 channels one would expect clear differences in the accumulation of Cav2 channel isoforms inside the synapse when indeed Syt11 is a specific trafficking partner of Cav2.2 channels.

The referee's suggestion to use pharmacological tools for demonstrating a reduction in presynaptic Cav2.2 channels in Syt11^{-/-} neurons is in agreement with the same recommendation made by referee 1 (Query 5). As recommended by referee 1, we assessed the relative contribution of Cav2.2 and Cav2.1 channels to synaptic release in cultured hippocampal neurons of Syt11^{+/+} and Syt11^{-/-} mice by recording the sEPSC frequencies in the presence of ω -conotoxin and ω -agatoxin. The data show a significant loss of presynaptic Cav2.2 channels in Syt11^{-/-} neurons and a partial compensatory increase in other Cav channels. We have incorporated these data into the Results and Discussion sections, presenting it in a new Figure 6. We also have included this finding into the "Highlights".

P1, I21

Syt11 knockout mice exhibit a deficit in presynaptic Cav2.2 channels and GABA_B receptors.

P8, I1

Hippocampal *Syt11*^{-/-} neurons exhibit a significant deficit in presynaptic Cav2.2 channels

Various types of Cav channels contribute to activity-dependent neurotransmitter release at brain synapses, with Cav2.1 and Cav2.2 channels being most prominent [55-58]. To determine the contribution of Cav2.2 channels to synaptic transmitter release in *Syt11*^{-/-} neurons and explore potential compensatory changes in Cav2.1 channels, we measured sEPSC frequencies in the consecutive presence of ω -conotoxin (blocking Cav2.2 channels), ω -conotoxin + ω -agatoxin (blocking Cav2.1 channels), and ω -conotoxin + ω -agatoxin + TTX (preventing action potential-dependent release) (Fig 6A). Blocking Cav2.2 channels by ω -conotoxin significantly reduced the frequency of sEPSCs and shifted the distribution of inter-event intervals toward larger values in both *Syt11*^{-/-} and *Syt11*^{+/+} neuronal cultures (Fig 6B and C). However, ω -conotoxin was significantly less efficient in inhibiting the sEPSC frequency in *Syt11*^{-/-} neurons compared to *Syt11*^{+/+} neurons (Fig 6D), consistent with a reduction in presynaptic Cav2.2 channels. Notably, the combined inhibitory effect of ω -agatoxin and TTX, but not of ω -agatoxin alone, was significantly larger in *Syt11*^{-/-} neurons when applied after ω -conotoxin (Fig 6D and E). Altogether, this indicates a reduction in the relative contribution of presynaptic Cav2.2 channels to synaptic release in *Syt11*^{-/-} neurons, partially offset by an upregulation of other Cav channels. Additionally, the lack of significant differences in TTX-insensitive mEPSCs between genotypes (Fig 6D) indicates comparable synapse density.

P9, I5.

However, subtype-selective Cav channel blockers revealed a significant reduction in presynaptic Cav2.2 channels in *Syt11*^{-/-} neurons, which are not fully compensated for by other Cav channels. This is consistent with the finding that Syt11 exhibits a preference for binding to Cav2.2 channels over Cav2.1 channels (this study), and that presynaptic slots accepting Cav2.2 channels reject Cav2.1 channels [57].

P14, I25.

Figure 6 - Cultured *Syt11*^{-/-} hippocampal neurons exhibit a deficit in presynaptic Cav2.2 channels

(A) Representative traces of sEPSCs from *Syt11*^{+/+} and *Syt11*^{-/-} hippocampal neurons in culture recorded at DIV15-19 in the presence of gabazine (10 μ M) before (control, black/red) and after application of ω -conotoxin (1 μ M, blue), ω -conotoxin + ω -agatoxin (500 nM, yellow) and ω -conotoxin + ω -agatoxin + TTX (1 μ M, magenta).

(B) Cumulative probability distributions of sEPSC inter-event intervals of *Syt11*^{+/+} and *Syt11*^{-/-} neurons recorded as in (A).

(C) Summary bar graph depicting the sEPSC frequency of *Syt11*^{+/+} and *Syt11*^{-/-} neurons recorded as in (A). In both genotypes, the frequency of sEPSCs was significantly reduced by the application of ω -conotoxin, ω -conotoxin + ω -agatoxin, and ω -conotoxin + ω -agatoxin + TTX. Wilcoxon test; α -values were adjusted by the Bonferroni correction for multiple comparisons. n = 9-11 neurons per genotype from 4 preparations.

(D) Summary bar graph depicting the percentage inhibition of sEPSC frequency by ω -conotoxin, ω -agatoxin, and TTX in cultured hippocampal neurons of *Syt11*^{+/+} (black) and *Syt11*^{-/-} mice (red). Inhibition by ω -conotoxin (blocking Cav2.2 channels) is significantly reduced in *Syt11*^{-/-} compared to *Syt11*^{+/+} neurons (*Syt11*^{+/+}: 60.55 \pm 4.42 % vs *Syt11*^{-/-}: 31.90 \pm 7.80 %, p = 0.005, Mann-Whitney U test). The ω -agatoxin-sensitive (blocking Cav2.1 channels) and TTX-sensitive components of inhibition, as well as the TTX-insensitive component (mEPSCs) show no significant difference between genotypes.

(E) The combined ω -agatoxin- and TTX-sensitive component of inhibition is significantly increased in *Syt11*^{-/-} compared to *Syt11*^{+/+} neurons (*Syt11*^{+/+}: 29.33 \pm 4.64 % vs *Syt11*^{-/-}: 47.78 \pm 7.37 %, p = 0.038, Mann-Whitney U test).

Values in bar graphs are presented as mean \pm SEM, n = 9-11 neurons per genotype from 4 preparations. Statistical significance is indicated as *p < 0.05, **p < 0.01, ns = not significant. Source data are available online for this figure.

Query 5. The experiments in Fig.4 J-L seem to suggest such a situation but also here I would recommend more controls. For example, the tonic activity of GABABR may be affected dependent on the composition of the signalling complex. May you can use specific antagonists for GABABR or GABA transporters as employed in previous studies (Laviv et al., 2010)?

It is indeed possible that the presence or absence of Syt11 in GBR complexes affects constitutive and/or tonic GBR activity. To investigate Syt11's impact on constitutive GBR activity, we conducted experiments in heterologous cells, enabling assays in the complete absence of ambient GABA. In the luciferase assay-reporter assay reporting $G\alpha$ signaling, Syt11 does not significantly change basal and GABA-induced luciferase activity, thus providing no evidence for effects on constitutive receptor activity (Fig 2A). Similarly, in the BRET assay monitoring $G\beta\gamma$ signaling in transfected HEK293T cells, we observed no constitutive GBR activity with CGP54626, an antagonist with inverse agonistic properties, irrespective of Syt11 presence (Fig 2B). In addition, we explored whether *Syt11*^{-/-} and *Syt11*^{+/+} neuronal cultures exhibit constitutive and/or tonic GBR activity. Recording sEPSCs in the presence of CGP54626 revealed no evidence of constitutive or tonic activity in either culture. We have included these data into a new Figure EV5

P4, I19.

Luciferase activity at baseline was not significantly different in the presence or absence of Syt11, supporting that Syt11 does not influence constitutive $G\alpha$ signaling of GBRs (Fig 2A).

P4, I25.

After blocking GBRs with 4 μ M CGP54626, an inverse agonist inhibiting constitutive GBR activity [34], the BRET signal returned to baseline without undershooting, irrespective of the presence of Syt11 (Fig 2B). The data from CGP54626 experiments corroborate that Syt11 does not induce constitutive receptor activity.

P7, I28.

We next tested whether the decreased release probability in *Syt11*^{-/-} neuronal cultures is due to an increase in the constitutive or tonic activity of GBRs. Application of the inverse agonist CGP54626 did not significantly change the baseline frequency of sEPSCs in either genotype (Fig EV5A and B), providing no evidence for constitutive or tonic GBR activity in our neuronal cultures. In addition, CGP54626 did not increase the frequency of sEPSCs above baseline when applied after baclofen (Fig EV5C).

P9, I2.

In both cultured *Syt11*^{-/-} neurons and heterologous cells expressing GBRs and KCTD16 without Syt11, we detected no increase in tonic or constitutive GBR activity, which could have explained the reduced release. However, subtype-selective Cav channel blockers revealed a significant reduction in presynaptic Cav2.2 channels in *Syt11*^{-/-} neurons, which are not fully compensated for by other Cav channels. This is consistent with the finding that Syt11 exhibits a preference for binding to Cav2.2 channels over Cav2.1 channels (this study), and that presynaptic slots accepting Cav2.2 channels reject Cav2.1 channels [57].

P11, I31.

Expression of Syt11 does not significantly change basal and GABA-induced luciferase activity. Non-linear regression curve fits of n = 8 independent experiments per condition. Mean \pm SEM, p = 0.5929, extra sum-of-squares F-test (middle). Baseline activity (BL), p = 0.5882, unpaired t-test (bottom).

(B) G β released upon GBR activation in HEK293T cells was monitored using a BRET assay reporting the binding of Venus-tagged G β to a membrane-associated GRK3ct-luciferase (top). Representative experiments (middle) and quantification of BRET changes (bottom) induced by the application of GABA and the inverse agonist CGP54626. Co-expression of Syt11 does not significantly alter GABA ($p = 0.2068$) and CGP54626 ($p = 0.9777$) induced BRET changes compared to control (unpaired Student's t test). Mean \pm SEM from $n = 8$ independent experiments recorded in triplicates, ns = not significant.

P17, I10.

Figure EV5 - Lack of tonic or constitutive GBR activity in cultured *Syt11^{+/+}* and *Syt11^{-/-}* hippocampal neurons

(A) Representative traces of sEPSCs recorded from a *Syt11^{+/+}* (top) and *Syt11^{-/-}* (bottom) neuron in the presence of gabazine (10 μ M) before (control, black/red) and after application of CGP54626 (4 μ M, yellow).

(B) Cumulative probability distributions of sEPSC inter-event intervals from *Syt11^{+/+}* (top) and *Syt11^{-/-}* (bottom) neurons recorded as in (A). In both genotypes, the sEPSC frequency (insets) was not significantly different in the presence of CGP54626 (CGP) compared to control (con). Upper inset: *Syt11^{+/+}* neurons (con: 8.16 ± 2.46 Hz vs CGP: 7.07 ± 1.83 Hz, $p = 0.438$). Lower inset: *Syt11^{-/-}* neurons (con: 5.74 ± 1.32 Hz vs bac: 6.31 ± 1.67 Hz, $p = 0.813$). Mean \pm SEM, ns = not significant, Wilcoxon matched-pairs signed rank test, $n = 5$ neurons per genotype from 3 preparations.

(C) Summary bar graph showing the sEPSC frequency of *Syt11^{+/+}* (left) and *Syt11^{-/-}* (right) neurons in the presence of gabazine (10 μ M) before (control, black/red) and after application of baclofen (100 μ M, blue) and baclofen + CGP54626 (4 μ M, yellow). Mean \pm SEM, **** $p < 0.0001$, ns = not significant, Friedman test and Dunn's multiple comparisons. *Syt11^{+/+}*, $n = 11$ neurons; *Syt11^{-/-}*, $n = 16$ neurons from 6 preparations.

Query 6. Based on such experiments you may can come up with a more direct interpretation of the bursting activity. As we know from several identified synapses there is not always a direct correlation between just the number of calcium channels and their impact on synaptic release probability see (Aldahabi et al., 2022; Eltes et al., 2017; Rebola et al., 2019). In summary, in my opinion the paper needs some further characterisation of the proposed functional consequences in addition to the subcellular identification of the complexes as mentioned above. The molecular interactions are confirmed several times and it would be interesting whether indeed GABABR together with Syt11 determine the population size and stability of Cav2.2 channels in individual synapses.

While Syt11 had previously been identified as a component of native GBR complexes, the specific binding partners of Syt11 within the GBR complex were unknown. This study marks the first demonstration of the molecular interactions between Syt11, Cav2.2 channels, and KCTD16. Additionally, we present evidence for the first time that Syt11 facilitates the assembly of GBR/Cav2.2 complexes by recruiting both signaling components to post-Golgi transport vesicles. In response to the referees' recommendations, additional experiments now reveal a significant reduction in presynaptic Cav2.2 channels in *Syt11^{-/-}* neurons. We found that this reduction cannot be compensated for by Cav2.1 channels, as they do not efficiently bind to Syt11 and fail to co-purify with Syt11 from brain tissue.

Dear Prof. Bettler

Thank you for the submission of your revised manuscript to EMBO reports. We have now received the full set of referee reports that is copied below. Both referees are very positive about the study and support publication.

Browsing through the manuscript myself, I noticed a few editorial things that we need before we can proceed with the official acceptance of your study.

- Please add up to 5 keywords.

- Please update the 'Conflict of interest' paragraph to our new 'Disclosure and competing interests statement'. For more information see

<https://www.embopress.org/page/journal/14693178/authorguide#conflictsofinterest>

- Please update the references to the alphabetical Harvard style. The abbreviation 'et al' should be used if more than 10 authors. You can download the respective EndNote file from our Guide to Authors

https://endnote.com/style_download/embo-reports/

- Appendix: even if it contains only one figure, we would still need a title page with a table of content and page numbers.

- Materials and Methods should be called Methods.

- The manuscript sections should be in the following order: Title page - Abstract & Keywords - Introduction - Results - Discussion - Methods - Data Availability - Acknowledgments - Disclosure Statement & Competing Interests - References - Figure Legends - Tables with legends - Expanded View Figure Legends.

- Please remove the 'Highlights' from the manuscript file. I have already captured these in the production forms. In addition to the bullet points, we would need a draft for a 1-2 sentence summary of the findings.

- Synopsis image: please provide it in either jpeg, TIFF or png format and its size needs to be 550 pixels wide x 200-600 pixels high.

- Please note that the specific URL for PXD044764 dataset should be provided in the data availability statement, i.e., we need a link that resolves directly to the dataset at PRIDE. Please do not forget to remove the reviewer access data.

- Please also remove the statement "Any additional information required...". The Data Availability section should only list information about datasets at public repositories.

- Our production/data editors have asked you to clarify several points in the figure legends (see below). Please incorporate these changes in the manuscript and return the revised file with tracked changes with your final manuscript submission.

A) Please note that a separate 'Data Information' section is required in the legends of figures 4c-e; 5b-c, e-h; 6c-e; EV 3b-c; EV 4a-c.

B) Please note that in figures 4e; 5b-c, e-h; there is a mismatch between the annotated p values in the figure legend and the annotated p values in the figure file that should be corrected. Only p-values that are actually shown in the figure panel(s) should (and must) be defined in the legends, all others should be removed from (or added to) the legend.

C) Please note that information related to n is missing in the legends of figure 4e; EV 2b.

D) Please note that the error bar is not defined in the legend of figure 3d.

E) Please note that the scale bar needs to be defined for figure EV 2a.

F) Please note that the white arrowheads are not defined in the legends of figures EV 4a, c. This needs to be rectified.

- On a different note, I would like to alert you that EMBO Press offers a new format for a video-synopsis of work published with us, which essentially is a short, author-generated film explaining the core findings in hand drawings, and, as we believe, can be very useful to increase visibility of the work. This has proven to offer a nice opportunity for exposure i.p. for the first author(s) of the study. Please see the following link for representative examples and their integration into the article web page:

<https://www.embopress.org/doi/full/10.15252/embo.2019103932>

With kind regards,

Referee #1:

The authors have responded well to my concerns. This is a very good paper which advances the field substantially.

Referee #3:

The authors have answered all my questions and provide additional data that support their conclusions. I have no further comment on the work and congratulate the authors for a very sound study.

All editorial and formatting issues were resolved by the authors.

Prof. Bernhard Bettler
University of Basel
Department of Biomedicine
Pharmazentrum
Klingelbergstrasse 5070
Basel CH-4056
Switzerland

Dear Prof. Bettler,

I am very pleased to accept your manuscript for publication in the next available issue of EMBO reports. Thank you for your contribution to our journal.

Kind regards,
